# Negative regulation of ABA signaling by WRKY33 is critical for *Arabidopsis* immunity towards *Botrytis cinerea* 2100

Shouan Liu[1†], Barbara Kracher[1], Jörg Ziegler[2], Rainer P Birkenbihl[1], Imre E Somssich[1]*

[1]Department of Plant Microbe Interactions, Max Planck Institute for Plant Breeding Research, Köln, Germany; [2]Department of Molecular Signal Processing, Leibniz Institute of Plant Biochemistry, Halle, Germany

**Abstract** The *Arabidopsis* mutant *wrky33* is highly susceptible to *Botrytis cinerea*. We identified >1680 *Botrytis*-induced WRKY33 binding sites associated with 1576 *Arabidopsis* genes. Transcriptional profiling defined 318 functional direct target genes at 14 hr post inoculation. Comparative analyses revealed that WRKY33 possesses dual functionality acting either as a repressor or as an activator in a promoter-context dependent manner. We confirmed known WRKY33 targets involved in hormone signaling and phytoalexin biosynthesis, but also uncovered a novel negative role of abscisic acid (ABA) in resistance towards *B. cinerea* 2100. The ABA biosynthesis genes *NCED3* and *NCED5* were identified as direct targets required for WRKY33-mediated resistance. Loss-of-WRKY33 function resulted in elevated ABA levels and genetic studies confirmed that WRKY33 acts upstream of *NCED3/NCED5* to negatively regulate ABA biosynthesis. This study provides the first detailed view of the genome-wide contribution of a specific plant transcription factor in modulating the transcriptional network associated with plant immunity.

*For correspondence: somssich@mpipz.mpg.de

Present address: †Tea Research Institute, Chinese Academy of Agricultural Sciences, Hangzhou, China

Competing interests: The authors declare that no competing interests exist.

## Introduction

Necrotrophic fungi including *Botrytis cinerea*, *Fusarium oxysporum*, and *Alternaria brassicicola* are the largest class of fungal phytopathogens causing serious crop losses worldwide (*Łaźniewska et al., 2010*). These pathogens extract nutrients from dead host cells by producing a variety of phytotoxic compounds and cell wall degrading enzymes (*Williamson et al., 2007*; *Mengiste, 2012*). *B. cinerea* has a broad host-range, causes pre- and postharvest disease, and is the second most agriculturally important fungal plant pathogen (*Dean et al., 2012*).

Plant immunity towards *B. cinerea* appears to be under complex poorly understood genetic control (*Rowe and Kliebenstein, 2008*). Apart from the *Arabidopsis thaliana RESISTANCE TO LEPTOS-PHAERIA MACULANS 3* (*RLM3*), no major *R-gene* has been associated with resistance to necrotrophs. However, over the past two decades numerous genes that influence the outcome of *B. cinerea*—host interactions have been identified (*Mengiste, 2012*). Among these are several transcription factors (TFs) consistent with the large transcriptional reprogramming observed in host cells upon *Botrytis* infection (*Birkenbihl and Somssich, 2011*; *Birkenbihl et al., 2012*; *Windram et al., 2012*). In *Arabidopsis*, several MYB-type TFs regulate distinct host transcriptional responses towards *B. cinerea*. BOS1 (*BOTRYTIS SUSCEPTIBLE 1*)/MYB108 appears to restrict necrosis triggered by *B. cinerea* and *A. brassicicola*, and loss-of-BOS1 function increased plant susceptibility (*Mengiste, 2012*). In response to stress and *B. cinerea* infection, BOS1 physically interacts with and is ubiquitinated by BOI, a RING E3 ligase that contributes to defense by restricting the extent of necrosis (*Luo et al., 2010*). MYB51 is involved in the transcriptional activation of indole glucosinolate biosynthetic genes,

**eLife digest** Crop yields can be badly affected by diseases caused by some fungi and other microbes. One fungus called *Botrytis cinerea* is able to infect many different species of crop plants—including tomatoes and grapes—and can cause severe damage both before and after harvest. This fungus belongs to a group of microbes that kill the plant cells they invade and then extract the nutrients from the dead cells.

Some plants are able to resist infection by *B. cinerea* and researchers have identified several proteins that are involved in this resistance. One such protein is called WRKY33, which is able to bind to DNA to regulate the activity of particular genes. However, it was not clear exactly which genes were involved in the response to *B. cinerea*.

*Arabidopsis thaliana* is a small flowering plant that is often used in research. Mutant *A. thaliana* plants lacking WRKY33 are very susceptible to infection with *B. cinerea*. Here, Liu et al. use several genetic techniques to find out which genes WRKY33 regulates when *A. thaliana* plants are exposed to the fungus. The experiments indicate that WRKY33 can alter the activity of over 300 genes. Some of these genes had previously been shown to be targets of WRKY33 and are involved in cell responses to plant hormones and the production of an antimicrobial molecule called camalexin.

Liu et al. also show that two genes called *NCED3* and *NCED5*—which are required for the production of a plant hormone called abscisic acid—are repressed by WRKY33. Mutant plants lacking WRKY33 had increased levels of abscisic acid and further experiments suggested that the repression of *NCED3* and *NCED5* by WRKY33 is important to resistance against the fungus.

Liu et al.'s findings provide the first detailed view of which genes in *A. thaliana* are regulated by WRKY33 when the plant is exposed to *B. cinerea*. A future challenge is to understand how blocking the production of abscisic acid protects plants against *B. cinerea* and other similar fungi.

which also contributes to resistance towards necrotrophs (*Kliebenstein et al., 2005*; *Sánchez-Vallet et al., 2010*). In contrast, the MYB-related genes *ASYMMETRIC LEAVES 1 (AS1)* and *MYB46* appear to play a role in disease susceptibility as such mutants show increased disease resistance towards *B. cinerea* (*Nurmberg et al., 2007*; *Ramírez et al., 2011*).

Ethylene and jasmonic acid (ET, JA) signaling are critical for host immunity to necrotrophic pathogens, and several transcriptional activators and repressors of the ET and JA pathways impact resistance to *B. cinerea* (*Glazebrook, 2005*; *Bari and Jones, 2009*). In particular the TFs ERF1, ORA59, ERF5, ERF6, and RAP2.2, have regulatory functions in host susceptibility to this fungus. (*Berrocal-Lobo et al., 2002*; *Pré et al., 2008*; *Moffat et al., 2012*; *Zhao et al., 2012*). Transgenic *Arabidopsis* lines overexpressing *ERF1* or *ORA59* confer resistance to *B. cinerea* (*Kazan and Manners, 2013*), whereas *RNAi-ORA59* silenced lines were more susceptible (*Berrocal-Lobo et al., 2002*; *Pré et al., 2008*). Both ERF1 and ORA59 appear to be the key integrators of the ET- and JA-signaling pathways (*Pieterse et al., 2009*). In contrast, the bHLH transcription factor MYC2/JIN1 is a master regulator of diverse JA-mediated responses by antagonistically regulating two distinct branches of the JA signaling pathway in response to necrotrophs (*Kazan and Manners, 2013*).

The WRKY family of TFs modulates numerous host immune responses (*Pandey and Somssich, 2009*). In particular, WRKY33 is a key positive regulator of host defense to both *A. brassicicola* and *B. cinerea* (*Zheng et al., 2006*; *Birkenbihl et al., 2012*). WRKY33 was directly phosphorylated in vivo by the MAP kinases MPK3 and MPK6 upon *B. cinerea* infection and subsequently activated *PAD3* expression by direct binding to its promoter (*Mao et al., 2011*). *PAD3* encodes a key biosynthetic enzyme required for the production of the antimicrobial compound camalexin. Moreover, WRKY33 directly interacted with its own promoter, suggesting a positive feedback regulatory loop on *WRKY33* expression. WRKY33 was also found to interact with the VQ-motif containing protein MAP KINASE SUBSTRATE1 (MKS1/VQ21) and to form a ternary complex with the MAP kinase MPK4 within the nucleus of resting cells (*Andreasson et al., 2005*; *Qiu et al., 2008*). Upon challenge with the hemibiotrophic pathogen *Pseudomonas syringae* or upon elicitation by the microbe-associated molecular pattern (MAMP) flg22, the active epitope of the bacterial flagella, activated MPK4 phosphorylates MKS1 thereby releasing WRKY33 from the complex and leading to its detection at the *PAD3* promoter.

We previously reported that activation of *Arabidopsis WRKY33* resulted in rapid and massive host transcriptional reprogramming upon infection with *B. cinerea* strain 2100 (*Birkenbihl et al., 2012*). Compared to resistant wild-type (WT) plants, susceptible *wrky33* mutants displayed early inappropriate activation of salicylic acid (SA)-related host responses, elevated SA and JA levels, and down-regulation of JA-associated responses at later infection stages. Consistent with these results ChIP analysis demonstrated that WRKY33 directly binds to the regulatory regions of *JAZ1* and *JAZ5*, two genes encoding repressors of JA signaling, but also to the ERF class TF gene *ORA59* involved in JA-ET crosstalk, and to two camalexin biosynthesis genes *CYP71A13* and *PAD3* (*Birkenbihl et al., 2012*). Although *pad3* plants are susceptible to *B. cinerea* 2100, *wrky33* mutants are more highly susceptible. Genetic studies revealed that altered SA responses at later infection stages may contribute to the susceptibility of *wrky33* to *B. cinerea*, but were insufficient for WRKY33-mediated resistance (*Birkenbihl et al., 2012*). Thus, WRKY33 apparently targets additional genes whose functions are critical for establishing full WRKY33-dependent resistance towards this necrotroph.

In this paper, we performed ChIP-seq and RNA-seq analyses to identify WRKY33-regulated target genes in the *A. thaliana* genome following infection with *B. cinerea* 2100. The study uncovered numerous targets many of which are associated with the regulation of hormonal signaling pathways. Expression of the majority of WRKY33 direct targets is down-regulated upon infection, but some notably genes of camalexin biosynthesis are strongly up-regulated, indicating that WRKY33 is a dual functional TF acting in a promoter-context dependent manner. Subsequent genetic and hormonal studies verified components of abscisic acid (ABA) biosynthesis as being critical for WRKY33-dependent resistance towards this necrotrophic fungus. This study provides the first genome-wide view of the gene regulatory network underlying plant immunity governed by a host specific TF.

## Results

### Genome-wide detection of *Arabidopsis* WRKY33 binding sites in response to *B. cinerea* 2100

To gain a deeper insight into how WRKY33 regulates plant immunity towards *B. cinerea* 2100, we performed ChIP-seq for genome-wide in vivo identification of WRKY33 DNA-binding sites. For this, a transgenic *wrky33* null mutant expressing an HA epitope-tagged *WRKY33* construct under the control of its native promoter ($P_{WRKY33}$:*WRKY33-HA*) was used. This line complemented the *B. cinerea* 2100 susceptibility phenotype of *wrky33* plants resulting in resistance similar to WT Col-0 plants (*Birkenbihl et al., 2012*). Rosette leaves of 4-week old plants, mock treated or spray inoculated with spores of *B. cinerea* 2100, were collected at 14 hr post inoculation and used to perform ChIP-seq. The 14 hr timepoint was selected based on the induced WRKY33-HA protein levels observed in western blots (*Figure 1A*). No WRKY33-HA protein was detected in the absence of infection. Besides the non-induced sample, we used identically treated WT plant tissue lacking *WRKY33-HA* as an additional negative control. Two biological replicates each were analyzed. The previously identified WRKY33 in vivo target genes, *CYP71A13* and *PAD3*, were used to monitor by ChIP-qPCR specific enrichment in samples used for library construction and sequencing (*Birkenbihl et al., 2012*).

We identified 1684 high confidence WRKY33 binding sites common to both replicates, which are associated with 1576 genes (*Figure 2C, Supplementary file 1*). WRKY33 binding to all detected sites was dependent on prior infection with *B. cinerea* 2100. Over 78% of the identified peak regions were located in promoter (1 kb region upstream of the transcription start site) or 5′ intergenic regions (*Figure 1B*) and 15.4% were located near transcription termination sites. Less than 1% and 5% of the peaks were located in exons and intronic regions, respectively (*Figure 1B*). The genome-wide local distribution of peak regions relative to genes showed clear accumulation of WRKY33 binding at about −300 bp from the transcription start sites (*Figure 1C*). The fidelity of the ChIP-seq data was subsequently confirmed by ChIP-qPCR for a number of genes (*Supplementary file 2*). Moreover, nearly all previously reported WRKY33 in vivo targets including *PAD3, CYP71A13, ACS2, JAZ1, ORA59, TRX-h5,* and *WRKY33* itself were successfully identified in our ChIP-seq dataset (*Mao et al., 2011; Birkenbihl et al., 2012; Li et al., 2012*).

Numerous studies have revealed that WRKY proteins specifically bind to a DNA motif, TTGACT/C, termed the W-box (*Rushton et al., 2010*), although adjacent bases (W-box extended motifs) can also influence binding (*Ciolkowski et al., 2008*). Using the DREME/MEME software, we determined conserved consensus sequences within high confidence WRKY33 binding sites across the genome.

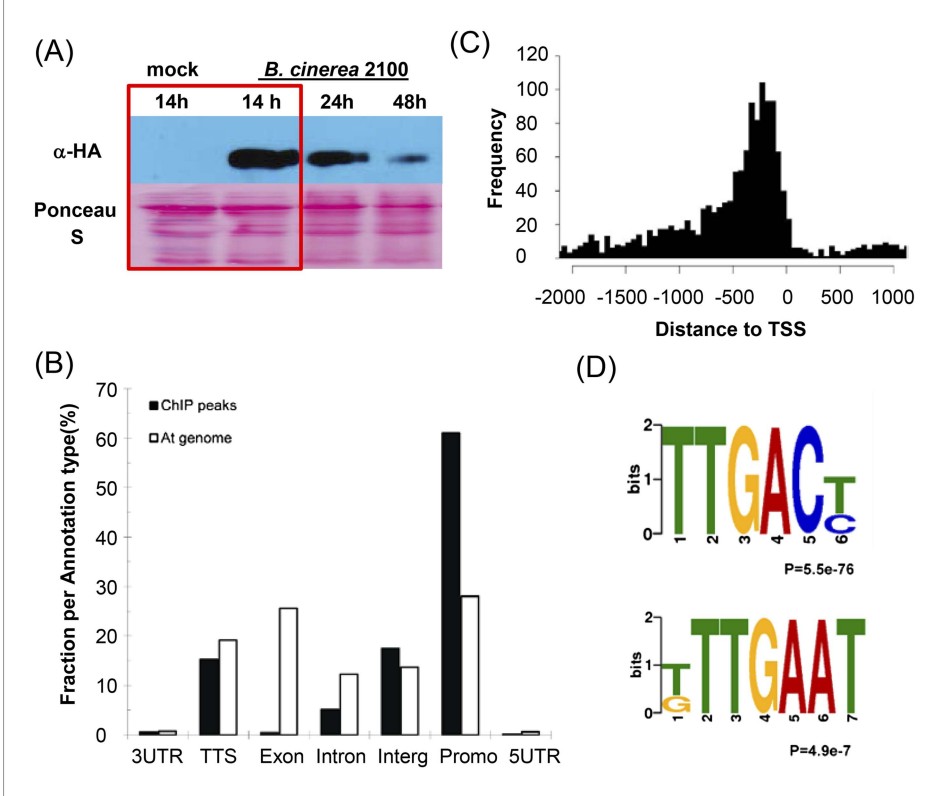

**Figure 1**. Genome-wide identification of *Arabidopsis* WRKY33 binding sites. (**A**) Western-blot analysis of WRKY33-HA protein levels after mock treatment or spray-inoculation of $P_{WRKY33}$:*WRKY33-HA* transgenic plants with *B. cinerea* 2100 spores. Plant material selected for ChIP-seq is boxed. (**B**) Relative binding-peak distribution across genomic regions. The 1 kb region upstream of the transcription start site is defined as promoter. The fraction of nucleotides in the complete At genome associated with each annotation type is included in the figure as background control (At genome). (**C**) Distribution of identified WRKY33 binding sites relative to the TSS. (**D**) Conserved DNA elements enriched within the 500 bp WRKY33 binding peak regions identified by DREME motif search. The TTGACT/C motif represents the well-established W-box, whereas T/GTTGAAT is an identified new motif.

The following figure supplements are available for figure 1:

**Figure supplement 1**. Conserved DNA elements within the 500 bp WRKY33 binding peak summit regions identified by MEME.

**Figure supplement 2**. WRKY33 does not bind to the G/TTTGAAT motif.

Of the 1684 identified WRKY33 binding regions, 80% contained the well-established W-box motif (*Figure 1D*). We also found W-box extended sequence motifs within the WRKY33 binding regions (*Figure 1—figure supplement 1A,B*). These W-box extended motifs also included the core sequence GACTTTT (*Figure 1—figure supplement 1C*), which was reported to be bound by *Arabidopsis* WRKY70 and to be required for WRKY70-activated gene expression (*Machens et al., 2014*).

Apart from the W-box and W-box variants, we found one additional sequence motif, T/GTTGAAT that occurs in 60% of the WRKY33 binding regions (*Figure 1D*). More than 48% (817 out of 1684) of WRKY33 binding peaks contained both this new motif and the W-box (*Figure 1—figure supplement 1D*). We performed electrophoresis mobility shift assays (EMSA) using recombinant WRKY33 protein to determine whether this newly identified DNA element is bound by WRKY33. Two DNA oligonucleotide probes were synthesized whose sequences were derived from two WRKY33 targets (*WAKL7*, *PROPEP3*) containing either one or three copies of the motif, respectively. A previously described DNA oligonucleotide containing three W-boxes (*Mao et al., 2011*) and a W-box mutated version hereof W-boxmut served as positive and negative controls. A clear interaction (mobility shift) was observed

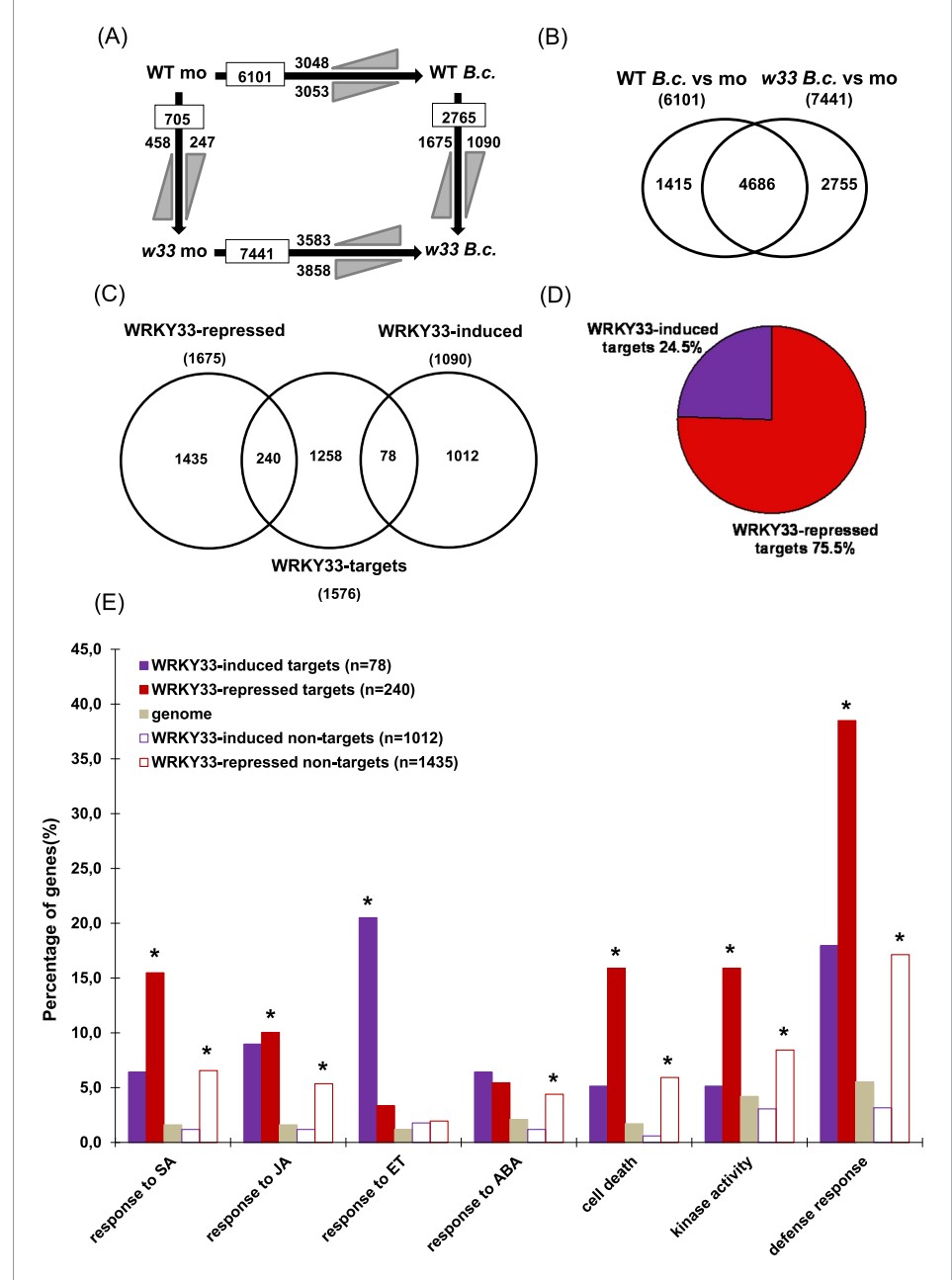

**Figure 2**. WRKY33-regulated direct target genes in response to **B**. *cinerea* 2100 infection. (**A**) Number of differentially expressed genes (≥ twofold; p ≤ 0.05) between WT and *wrky33* (*w33*) at 14 hr after mock treatment (mo) or spray inoculation with spores of *B. cinerea* 2100 (*B.c.*) identified by RNA-seq. Indicated are total numbers (boxed) and numbers of up-regulated ( ) and down-regulated genes ( ) between treatments or genotypes. (**B**) Venn diagram illustrating the total numbers and the number of common genes affected in WT and *wrky33* 14 hr post *B. cinerea* 2100 inoculation. (**C**) Venn diagram showing the numbers of genes common to WRKY33-regulated genes and WRKY33 target genes. (**D**) Percentage of WRKY33-repressed and WRKY33-induced target genes (in total 318). (**E**) Enrichment of specific Gene Ontology (GO) terms related to defense response, kinase activity, cell death, and hormone responses among WRKY33-regulated target and non-target genes (compared to the overall genome). The y-axis indicates the percentages of genes associated to each GO category in each gene set. Asterisks indicate significant enrichment (adj. p value < 0.05) of genes associated to the respective GO term within a gene set as determined by GO term

*Figure 2. continued on next page*

*Figure 2. Continued*

enrichment analysis with goseq (null distribution approximated as Wallenius distribution; correction for potential count biases via probability weighting).
The following figure supplements are available for figure 2:

**Figure supplement 1**. GO graph visualization of top GO terms enriched among WRKY33-regulated WRKY33 target genes.
**Figure supplement 2**. Analysis of WRKY33-regulated target genes associated to GO-terms 'hormone responses and cell death'.

between WRKY33 and the labeled W-box probe but not with W-boxmut and the two probes harboring the T/GTTGAAT motifs (M-3 and M-7; *Figure 1—figure supplement 2*). Specificity of W-box binding was confirmed in competition experiments, wherein only an excess of the W-box probe was able to compete for binding of the protein. Thus, T/GTTGAAT does not appear to be a WRKY33 direct binding site and its functionality remains unclear.

## Differential expression of WRKY33 target genes upon *B. cinerea* 2100 infection

ChIP-seq studies in different organisms have revealed that the majority of binding sites bound by specific TFs in vivo do not result in altered expression levels of associated genes (*MacQuarrie et al., 2011*; *Chang et al., 2013*; *Fan et al., 2014*). To investigate the impact of WRKY33 binding on target gene expression, we performed RNA-seq and examined *WRKY33*-mediated gene expression changes in mock and *B. cinerea* 2100 (14 hpi) treated 4-week old *wrky33* and WT plants. Three independent biological replicates were generated and analyzed allowing us to identify genes with consistently altered expression after inoculation. In WT plants, the expression of 6101 genes was altered twofold or more ($p \leq 0.05$) compared to non-infected plants, with 3048 genes being up-regulated and 3053 genes being down-regulated (*Figure 2A*). In *wrky33*, upon infection, the expression of 7441 genes was altered more than twofold, 3583 of them being up-regulated and 3858 down-regulated. A common set of 4686 genes showed changes upon infection in both genotypes (*Figure 2A,B*). Comparative profiling of mock treated plants identified 705 genes that were differentially expressed between *wrky33* and WT in the absence of the pathogen, 458 of them being up-regulated and 247 down-regulated (*Figure 2A*). Comparing the expression profiles of *B. cinerea* infected *wrky33* and WT plants (*wrky33 B.c* vs WT *B.c*), we identified 2765 differentially expressed genes dependent on WRKY33, of which 1675 were up-regulated in the mutant (termed WRKY33-repressed genes) and 1090 were down-regulated in the mutant (termed WRKY33-induced genes; *Figure 2A,C*).

We then compared the WRKY33-dependent differentially expressed gene set obtained by RNA-seq with the WRKY33 target gene set revealed by ChIP-seq. This comparison identified 318 WRKY33-regulated target genes that were both bound by WRKY33 and exhibited WRKY33-dependent altered gene expression (*Figure 2C*). Of these, 240 (75%) were repressed upon infection while 78 (25%) were induced (*Figure 2C,D*). We named those genes WRKY33-repressed targets and WRKY33-induced targets, respectively. Based on this analysis, WRKY33 appears to have a prominent repressive role on the transcription of many specific host genes indicating a negative regulatory function of WRKY33 in mediating immunity to this pathogen. Genes displaying altered expression in the *wrky33* mutant compared to WT but showing no binding of WRKY33 at their respective gene loci were defined as WRKY33-dependent non-targets (1435 WRKY33-repressed non-targets and 1012 WRKY33-induced non-targets; *Figure 2C*). The overlap between observed WRKY33 binding and altered expression of the associated genes upon fungal infection was around 20% (318 of 1576). This fraction is similar to values reported for other plant TFs such as EIN3, HBI1, and BES1 (*Yu et al., 2011*; *Chang et al., 2013*; *Fan et al., 2014*).

## WRKY33 represses transcription of many plant immunity genes

Compared to the entire genome the identified WRKY33-regulated targets were significantly enriched in gene ontology (GO) categories involved in diverse biological processes and molecular functions

related to different forms of stress, external and endogenous stimuli, signal transduction, transport, metabolic processes and catalytic activity (p < 0.05; *Figure 2—figure supplement 1*), and many of these genes are repressed upon *B. cinerea* infection (*Figure 2E*). For example, genes related to 'defense response' were highly overrepresented among WRKY33-repressed targets (38%) and in the WRKY33-repressed non-target sets (17%), suggesting that WRKY33 mainly functions as a repressor of plant defense responses. However, it is important to note that nearly 18% of the WRKY33-induced targets were associated to defense responses compared to only 3% of the WRKY33-induced non-targets. This indicates that WRKY33 can also act as a direct activator of defense gene expression, very likely in a promoter-context dependent manner. Particularly prominent among the WRKY33-induced targets are genes associated with responses to the phytohormone ethylene (ET; 21%).

Apart from hormonal pathways discussed below, genes associated with the GO terms 'cell death' or related to diverse 'kinase activities' were markedly enriched among WRKY33-repressed targets and non-targets (*Figure 2E*). 42 out of 318 WRKY33-regulated targets are involved in cell death, and 38 of these appear to be repressed by WRKY33 (*Supplementary file 3*). This WRKY33-mediated repression may be an important feature required to reinforce resistance towards the necrotroph *B. cinerea* that depends on dead host tissue to complete its life cycle. Furthermore, 41 of the WRKY33-regulated target genes encode for various kinases, and again the majority of these genes appear to be negatively regulated by WRKY33 (*Supplementary file 4*). For the WRKY33-regulated target *LecRK VI.2*, a critical role in resistance against hemibiotrophic *P. syringae* pv. tomato DC3000 and necrotrophic *Pectobacterium carotovorum* bacteria has been demonstrated (*Singh et al., 2012*; *Huang et al., 2014*).

Several TF gene families involved in defense were targeted by WRKY33. In total, WRKY33 binding was found at 133 TF gene loci. Predominant among these are members of the AP2/ERFs, MYBs, WRKYs, and NACs families (*Figure 3A*). However, expression of only 16% (21 of 133) of these genes was directly modulated in a WRKY33-dependent manner after *B. cinerea* infection (complete list see *Supplementary file 5*). WRKY factors are predicted to form a highly interconnected regulatory sub-network (*Llorca et al., 2014*). Indeed, 18 *WRKY* genes were identified as direct targets of WRKY33 (*Figure 3A*). However, only seven genes, *WRKY33, WRKY38, WRKY41, WRK48, WRKY50, WRKY53,* and *WRKY55,* showed altered expression upon WRKY33 binding at 14 or 24 hpi (*Figure 3B–G*; *Figure 3—figure supplement 1, Supplementary file 5*). Binding to the *WRKY33* promoter is consistent with reports suggesting a positive autoregulatory feedback loop resulting in high-level accumulation of WRKY33 in response to *B. cinerea* (*Mao et al., 2011*). In addition, WRKY33-dependent altered transcription of 18 other *WRKY* genes with no detectable WRKY33 binding was observed following fungal infection, indicating that these genes are indirectly regulated by WRKY33 (WRKY33-regulated non-targets; *Figure 3A*). WRKY33 function negatively affected expression of most of these *WRKY* targets.

## WRKY33 modulates transcription of genes associated with hormonal pathways

Genes encoding components of pathways related to the key phytohormone signaling molecules SA, JA, ET, and ABA were highly enriched in the WRKY33-regulated gene set (*Figure 2E*). Genes involved in SA response were overrepresented in WRKY33-repressed targets and non-targets (*Figure 2E*). This is consistent with our previous transcriptomic profiling showing that WRKY33 directly or indirectly repressed the expression of genes in SA biosynthesis and SA-mediated signaling (*Birkenbihl et al., 2012*). More than 80% (34 out of 42) of the SA-response targets are also associated with the GO term 'cell death' (*Figure 2—figure supplement 2A*), suggesting that WRKY33 repression of the SA pathway is linked to modulation of host cell death responses.

In contrast to SA signaling genes, genes responsive to ET were highly enriched in the WRKY33-induced target dataset, among them are *ACS6, ORA59,* and *ERF5* (*Figure 2E*). ACS6 is involved in *Botrytis*-induced ethylene production and plays an important role in plant immunity (*Han et al., 2010*; *Li et al., 2012*). ORA59 and ERF5 belong to the AP2/ERF TF family with ORA59 acting as an integrator of JA and ET signaling and as a positive regulator of resistance against *B. cinerea*, while ERF5 also regulates ET signaling and is a key component of chitin-mediated immunity (*Pré et al., 2008*; *Moffat et al., 2012*). Genes responsive to JA and ABA were also overrepresented in our GO term analysis, but in this case similar fractions of genes were identified among WRKY33-induced targets,

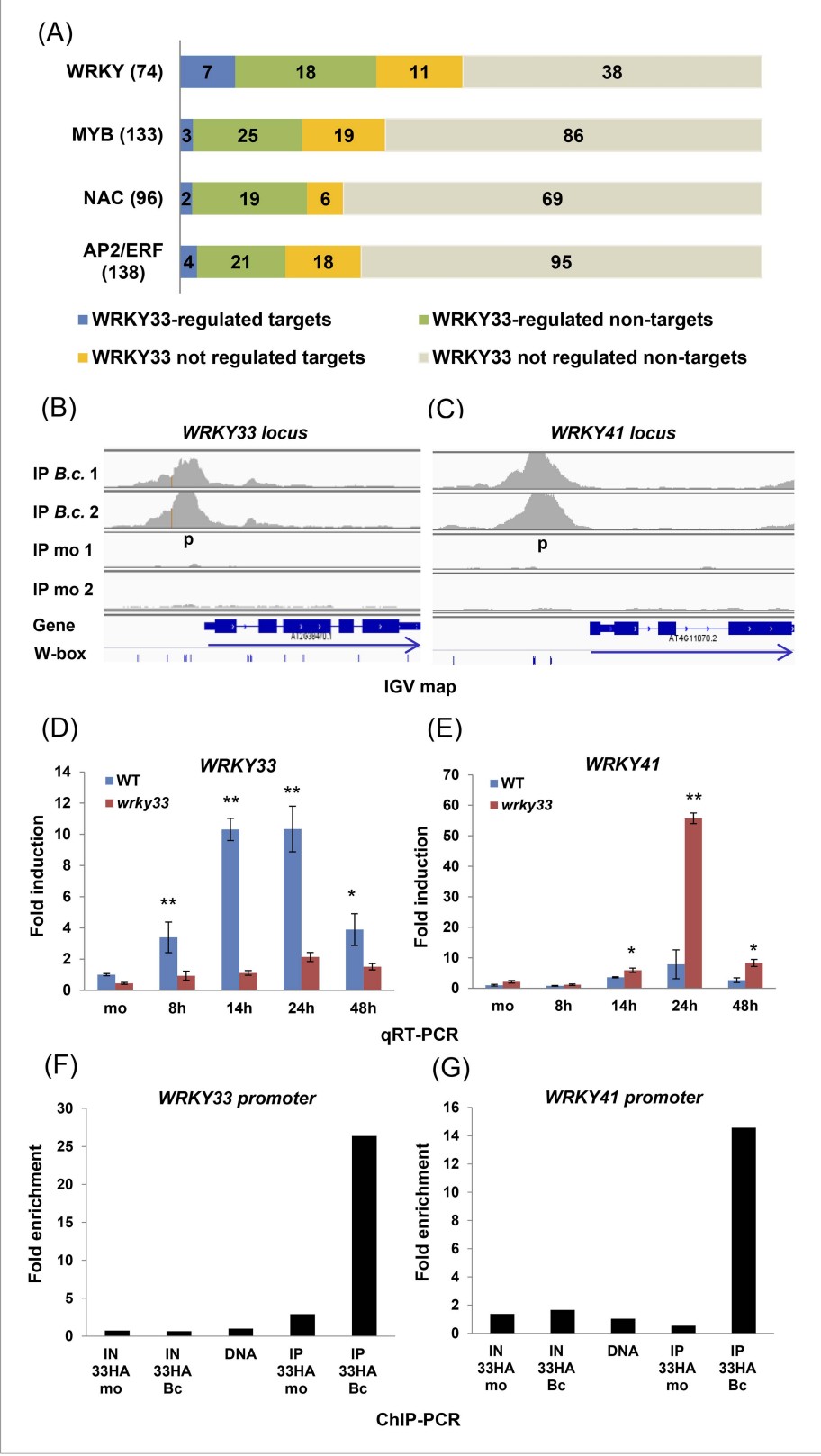

**Figure 3**. WRKY33-regulated transcription factor families commonly associated with stress responses. (**A**) WRKY, MYB, NAC, and AP2/ERF TF family genes are dominant targets of WRKY33 after *B. cinerea* 2100 infection. The total number of members for each TF family is given in parenthesis next to name. The number of WRKY33 directly or indirectly regulated family members are indicated. (**B**, **C**) Integrative Genomics Viewer (IGV) images of ChIP-seq data

*Figure 3. continued on next page*

*Figure 3. Continued*

revealing high infection-dependent WRKY33 binding at the promoters of *Arabidopsis WRKY33* (**B**) and *WRKY41* (**C**). Images for mock and *B.c.* treatment of both biological repetitions are shown (1 and 2). Structure of the targeted genes is indicated below along with the position of all W-box motifs within the loci. Arrows indicate direction of transcription. (**D**, **E**) qRT-PCR analysis of *B. cinerea* 2100-induced expression of *WRKY33* (**D**) and *WRKY41* (**E**) in WT and *wrky33* mutant plants at indicated time points post fungal spore application. All data were normalized to the expression of At4g26410 and fold induction values of all genes were calculated relative to the expression level of mock treated (mo) WT plants set to 1. Error bars represent SD of three biological replicates. Asterisks indicate significant differences between WT and *wrky33* (*, $p < 0.05$; **, $p < 0.001$; two-tailed *t*-test). (**F**, **G**) Validation of ChIP-seq data by ChIP-qPCR showing WRKY33 binding to its own promoter region (**F**) and to the *WRKY41* promoter (**G**). WRKY33-HA (33HA) plants were spray inoculated with spores of *B. cinerea* 2100 (Bc) or mock treated (mo) for 14 hr. Input DNA before immune precipitation (IN) and immune-precipitated DNA using an anti-HA antibody (IP) was analyzed by qPCR employing gene-specific primer pairs (p) indicated in the IGV graph. Shown is the fold enrichment of bound DNA relative to a non-bound DNA fragment from At2g04450. As a control for primer efficiency purified genomic DNA was included in the analysis. Each ChIP experiment was repeated at least twice with similar results.

The following figure supplement is available for figure 3:

**Figure supplement 1**. Validation of WRKY33 directly regulated *WRKY* genes.

---

WRKY33-repressed targets and WRKY33-repressed non-targets (*Figure 2E*). Some WRKY33-regulated targets were associated to more than one hormone response (*Figure 2—figure supplement 2B*), suggesting the involvement of WRKY33 in hormonal co-regulation or crosstalk.

In conclusion, our global analysis revealed that WRKY33 influences various hormonal responses upon infection with *B. cinerea* 2100, and that WRKY33 had both a positive and a negative functional relationship with a fraction of its direct targets.

## WRKY33-dependent resistance to *B. cinerea* 2100 involves ABA

Our previous genetic analyses excluded a major role of SA, JA and ET signaling in WRY33-dependent resistance towards *B. cinerea* 2100 (*Birkenbihl et al., 2012*). Here, two additional genes, *GH3.2* and *GH3.3*, encoding acyl-acid-amide synthetases capable of conjugating amino acids to JA and auxin were identified as being WRKY33-repressed targets (*Figure 8—figure supplement 3A*). GH3.3 controls JA homeostasis in seedlings, and *gh3.2* mutants showed increased resistance to *B. cinerea* (*González-Lamothe et al., 2012*; *Gutierrez et al., 2012*). Thus, we generated *wrky33 gh3.2 gh3.3* triple mutant plants but did not observe restoration of WT-like resistance towards *B. cinerea* 2100, indicating that they are not critical for WRKY33-dependent defense against this fungal strain (*Figure 8—figure supplement 3B*).

Interestingly, our global binding studies also revealed that WRKY33 binds to the promoter region of *NCED3* and to the 3′UTR region of *NCED5* (*Figure 4A,D*), two major genes encoding 9-cis-epoxycarotenoid dioxygenase, a key enzyme in the biosynthesis of ABA (*Leng et al., 2014*). The precise role of ABA in host defense remains enigmatic and ABA can positively or negatively impact the outcome of plant–microbe interactions, depending on the pathogens' lifestyle (*Robert-Seilaniantz et al., 2011*). WRKY33 binding to both gene loci was confirmed by ChIP-qPCR (*Figure 4C,F*). Transcript levels of *NCED3* and *NCED5* both increased in the *wkry33* mutant upon *B. cinerea* infection suggesting direct negative regulation by WRKY33 (*Figure 4B,E*). WRKY33 also bound to the *CYP707A3* promoter, a gene involved in ABA catabolism (*Leng et al., 2014*), but its expression decreased in the *wrky33* mutant and increased in WT plants after infection indicative of positive regulation by WRKY33 (*Figure 4G–I*).

These results suggest that WRKY33 represses ABA levels during *B. cinerea* 2100 infection, and that this repressor function is an important component in host resistance to this pathogen.

## Negative regulation of *NCED3* and *NCED5* by WRKY33 contributes to resistance to *B. cinerea* 2100

To clarify the involvement of ABA in WRKY33-mediated host defense to *B. cinerea* 2100, we analyzed ABA mutants with respect to their phenotypes after fungal infection. Previous reports have shown that

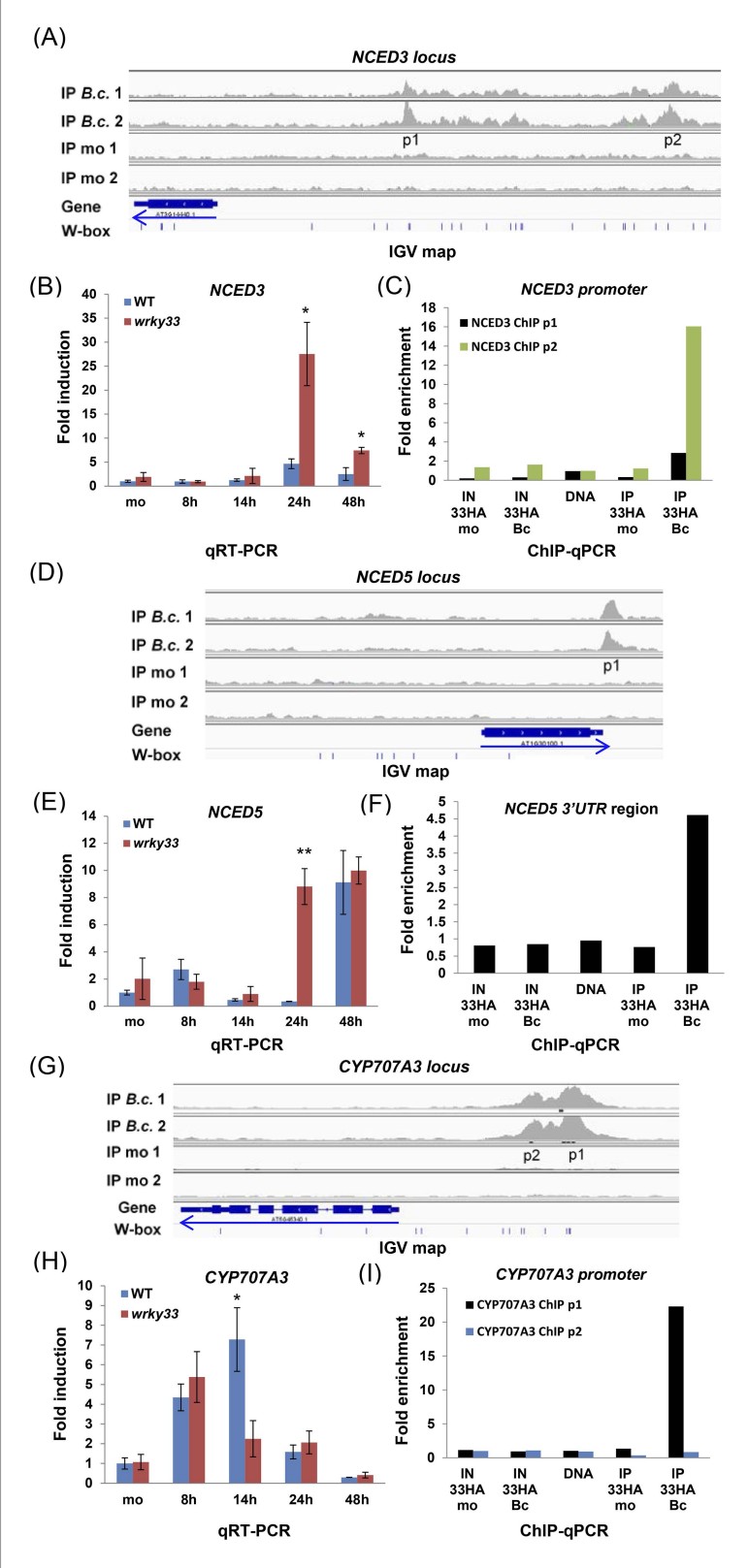

**Figure 4**. WRKY33 directly regulates target genes encoding ABA biosynthetic (*NCED3, NCED5*) and metabolic (*CYP707A3*) enzymes by binding to their promoters or 3'UTR after *B. cinerea* 2100 treatment. (**A**, **D**, **G**) IGV visualization of ChIP-seq data revealing infection-dependent WRKY33 enrichment at the *Arabidopsis NCED3* (**A**), *NCED5* (**D**) and *CYP707A3* (**G**) loci with features described in *Figure 3*. (**B**, **E**, **H**) qRT-PCR analysis of *B. cinerea*

*Figure 4. continued on next page*

*Figure 4. Continued*

2100-induced expression of *NCED3* (**B**) *NCED5* (**E**) and *CYP707A3* (**H**) in WT and *wrky33* as described in *Figure 3*.
(**C**, **F**, **I**) Validation of the ChIP-seq data for WRKY33 binding to the promoters of *NCED3* (**C**, with primer pairs p1 and p2)
and *CYP707A3* (**I**), and to the 3′UTR of *NCED5* (**F**) performed as described the legend to *Figure 3*.

---

*aba2-12* (*Adie et al., 2007*), *aba3-1* (*Léon-Kloosterziel et al., 1996*), and *nced3 nced5* (*Frey et al., 2012*) accumulated much less ABA than WT plants. Indeed, the *aba2-12*, *aba3-1*, *nced3-2*, *nced5-2*, and *nced3 nced5* mutants were nearly as resistant as WT plants to *B. cinerea* 2100 (*Figure 5B*; *Figure 5—figure supplement 1*). To test whether WT resistance towards this necrotroph is due to WRKY33-mediated repression of *NCED3* and *NCED5* expression we generated *wrky33 nced3*, *wrky33 nced5* double, and *wrky33 nced3 nced5* triple mutants, and tested their infection phenotypes.

ABA deficiency severely affects plant growth leading to stunted phenotypes, as observed in *nced3 nced5*, *aba2-12* and *aba3-1* mutants. A strong reduction of rosette diameter was also observed in the *wrky33 nced3 nced5* mutant under short day conditions (*Figure 5A*). However, unlike the *wrky33* mutant, the *wrky33 nced3 nced5* triple mutant showed clear resistance to *B. cinerea* 2100 similar to WT plants (*Figure 5B*). In contrast, the *wrky33 nced3* and *wrky33 nced5* double mutants were as susceptible as *wrky33* (*Figure 5—figure supplement 2*). Consistent with the observed resistance phenotype, qPCR analysis revealed strongly reduced fungal biomass in *wrky33 nced3 nced5* compared to *wrky33* plants at 3 days post infection (*Figure 5C*). This clearly indicates that increased expression levels of *NCED3* and *NCED5* in the *wrky33* mutant contribute to susceptibility toward *B. cinerea* 2100, and that a key function of WRKY33 in host immunity towards this pathogen is to repress ABA biosynthesis.

Since *nced3 nced5* mutants have reduced ABA levels, we tested whether exogenous application of ABA to the mutants could revert the resistant phenotype. Indeed, application of ABA together with the fungal spore droplet to leaves of the *wrky33 nced3 nced5* triple mutant partially rendered plants susceptible to *B. cinerea* 2100 (*Figure 5D*). Similar tests on WT plants did not alter host resistance (*Figure 5—figure supplement 3*).

*CYP707A* mutants affecting ABA metabolism were reported to accumulate more ABA than lines overexpressing ABA biosynthetic genes (*Finkelstein, 2013*). We therefore also tested the phenotypes of *cyp707a1*, *cyp707a2*, and *cyp707a3* following infection. Interestingly, all of these mutants remained as resistant as WT plants towards the fungus (*Figure 5—figure supplement 4*).

Taken together, our genetic analysis combined with our ChIP-seq and expression results strongly suggests that WRKY33-mediated control of *NCED3* and *NCED5* expression plays a critical role in host resistance towards *B. cinerea* 2100.

## WRKY33 controls hormone homeostasis in response to *B. cinerea*

Hormonal signaling appears to be affected in the susceptible *wrky33* mutant compared with the resistant WT (*Figure 2E*), and *wrky33 nced3 nced5* triple mutants restore WT-like resistance. Thus, we hypothesized that WRKY33 plays a critical regulatory role in hormone homeostasis. To test this, we measured hormonal levels in WT, *wrky33*, *nced3 nced5*, and *wrky33 nced3 nced5* plants following infection with *B. cinerea* 2100 at various time points. It is important to note that we have previously shown that up to 40 to 48 hpi no differences in fungal biomass, hyphal expansion, or other phenotypic criteria were observed between resistant WT and susceptible *wrky33* plants (*Birkenbihl et al., 2012*).

As expected, ABA and SA levels increased strongly in susceptible *wrky33* compared to resistant WT plants during fungal infection (*Figure 6A,B*). However, JA and ACC (precursor of ET) levels also increased strongly in *wrky33* compared to WT plants (*Figure 6C,D*). Interestingly, the elevated SA levels observed in *wkry33* appear to be a direct consequence of increased ABA levels as SA levels were clearly reduced in the resistant ABA-deficient *wrky33 nced3 nced5* compared to *wrky33* plants post infection. Moreover, the levels of JA and ACC were also reduced in *wrky33 nced3 nced5* at later infection stages. This suggests that ABA signaling exerts a positive role on the biosynthesis of these other hormonal components.

Taken together, our data indicate that a key function of WRKY33 in *B. cinerea* strain 2100 challenged WT plants is to limit ABA levels. Loss of *WRKY33* function affects hormonal homeostasis in the plant during infection, leading to elevated ABA activity and subsequently resulting in altered hormone signaling.

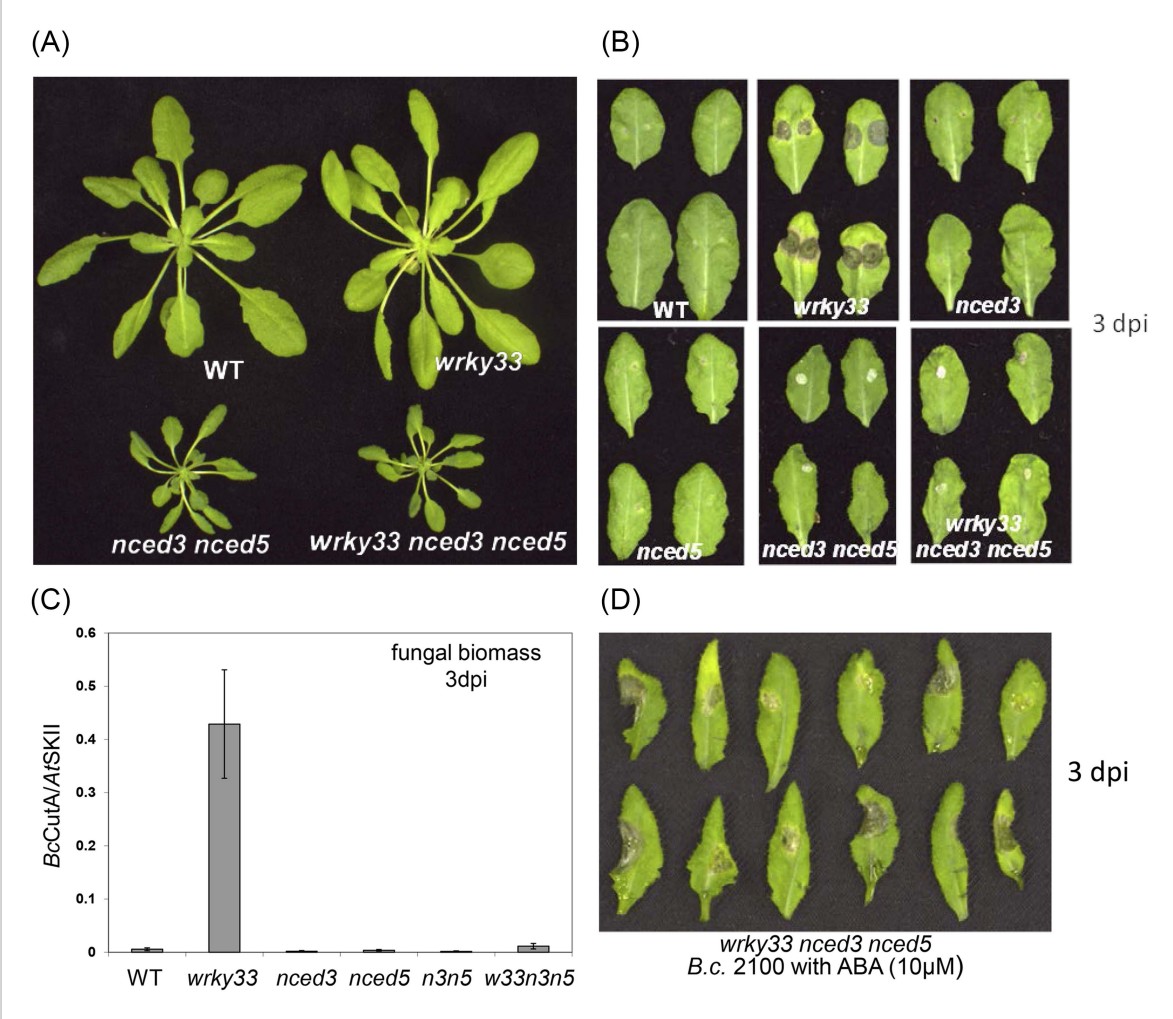

**Figure 5**. WRKY33 controls ABA-mediated plant susceptibility to *B. cinerea* 2100. (**A**) Growth phenotypes of WT, *wrky33*, *nced3 nced5*, and *wrky33 nced3 nced5 Arabidopsis* plants at 4 weeks under short day conditions. (**B**) *B. cinerea* infection phenotypes 3 days post inoculation of WT, *wrky33*, *nced3*, *nced5*, *nced3 nced5*, and *wrky33 nced3 nced5*. (**C**) *B. cinerea* biomass quantification on indicated *Arabidopsis* genotypes. For fungal biomass determination, the relative abundance of *B. cinerea* and *Arabidopsis* DNA was determined by qPCR employing specific primers for *Bc*Cutinase A and *At*SKII, respectively. (**D**) Exogenous application of ABA (10 μM) directly to infection droplets on *wrky33 nced3 nced5* leaves partially rendered plants susceptible to *B. cinerea* 2100. Upon completion of the infection experiments (3 dpi), leaves were detached and photographed. For the infections, one or two 2 μl droplets containing $2.5 \times 10^5$ spores were applied to each leaf.

The following figure supplements are available for figure 5:

**Figure supplement 1**. *B. cinerea* 2100 infection phenotypes of mature 4-week old leaves derived from *aba2-12* and *aba3-1* mutant plants.

**Figure supplement 2**. *B. cinerea* 2100 infection phenotypes of mature 4-week old leaves derived from *wrky33 nced3* and *wrky33 nced5* mutant plants.

**Figure supplement 3**. Phenotype of WT Col-0 plants treated with ABA.

**Figure supplement 4**. *B. cinerea* 2100 infection phenotypes of mature 4-week old leaves derived from cyp707a1, cyp707a2 and cyp707a3 mutant plants.

## In *wrky33 nced3 nced5* plants, expression of many up-regulated genes in *wrky33* is restored to WT-like levels

Over 75% of WRKY33-regulated target genes showed elevated expression in the susceptible *wrky33* mutant after *B. cinerea* 2100 infection (*Figure 2D*). To test these genes for altered expression in resistant *wrky33 nced3 nced5* plants, we performed qRT-PCR analyses. The transcript levels of several

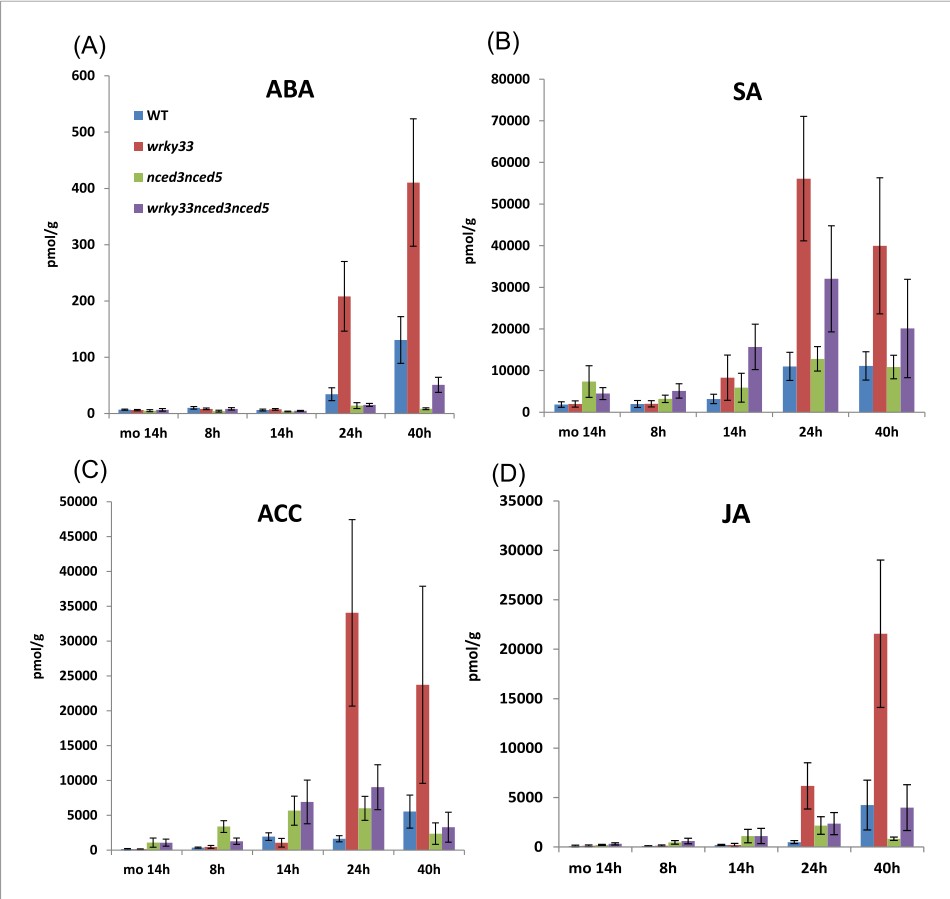

**Figure 6**. Hormone levels in different genotypes during *B. cinerea* 2100 infection. Concentrations of the hormones ABA (**A**), SA (**B**), ACC (**C**), and JA (**D**) were measured at 8, 14, 24, and 40 hpi in leaves of indicated *Arabidopsis* genotypes spray inoculated with spores of *B. cinerea* 2100. Mock treated plants (mo, 14 hr) served as a control. The data show the average values and SDs of the combined data from three independent experiments with up to four replicates each.

highly expressed SA-related genes observed in *wrky33* at 24 hpi decreased in the *wrky33 nced3 nced5* plants, often returning to WT states. These included: *ICS1*, *NPR1*, *NPR3*, *NPR4*, *TRX-h5*, and *FMO1* (*Figure 7*). However, not all SA-related genes were similarly affected as illustrated for *EDS1*, *PAD4*, *NIMIN1*, *PR1*, and *PR2*, whose expression levels remained significantly higher than in WT (*Figure 7*). These results imply that simultaneous mutations of *NCED3* and *NCED5* in the *wrky33* genotype partially impair SA biosynthesis and signaling.

Reduced transcript levels were also observed for other genes in the *wrky33 nced3 nced5* mutant at 24 hpi such as the TF genes *ERF1*, *NAC019*, *NAC055*, *NAC061*, *NAC090*, *WRKY41*, *WRKY48*, *WRKY53*, and *WRKY55*, indicating a positive effect of ABA on their expression (*Figure 7*). In contrast, expression of *WRKY38* and *WRKY50* increased in *wrky33 nced3 nced5* plants to even higher levels than observed in *wrky33* (*Figure 7*) suggesting a negative effect of ABA on these genes.

The *Botrytis*-induced expression of other ABA response genes including *ACS2*, *BIR1*, *CDPK1*, *MPK11*, and *CRK36* was also restored to WT levels in the *wrky33 nced3 nced5* mutant (*Figure 7*). Interestingly, *CDPK1*, *MPK11*, *CRK36*, and *NPR3* are also responsive to SA and these genes are associated with the GO term 'cell death', suggesting that ABA has a positive effect on cell death responses (*Figure 2—figure supplement 2A*, *Supplementary files 3, 4*).

In summary, WRKY33 suppresses the expression of many of its target genes by negatively regulating ABA responses. The subset of genes showing restoration of WT-like expression levels in the resistant triple mutant constitutes prime candidates whose functions may be causal for WRKY33-mediated resistance against this necrotrophic fungus.

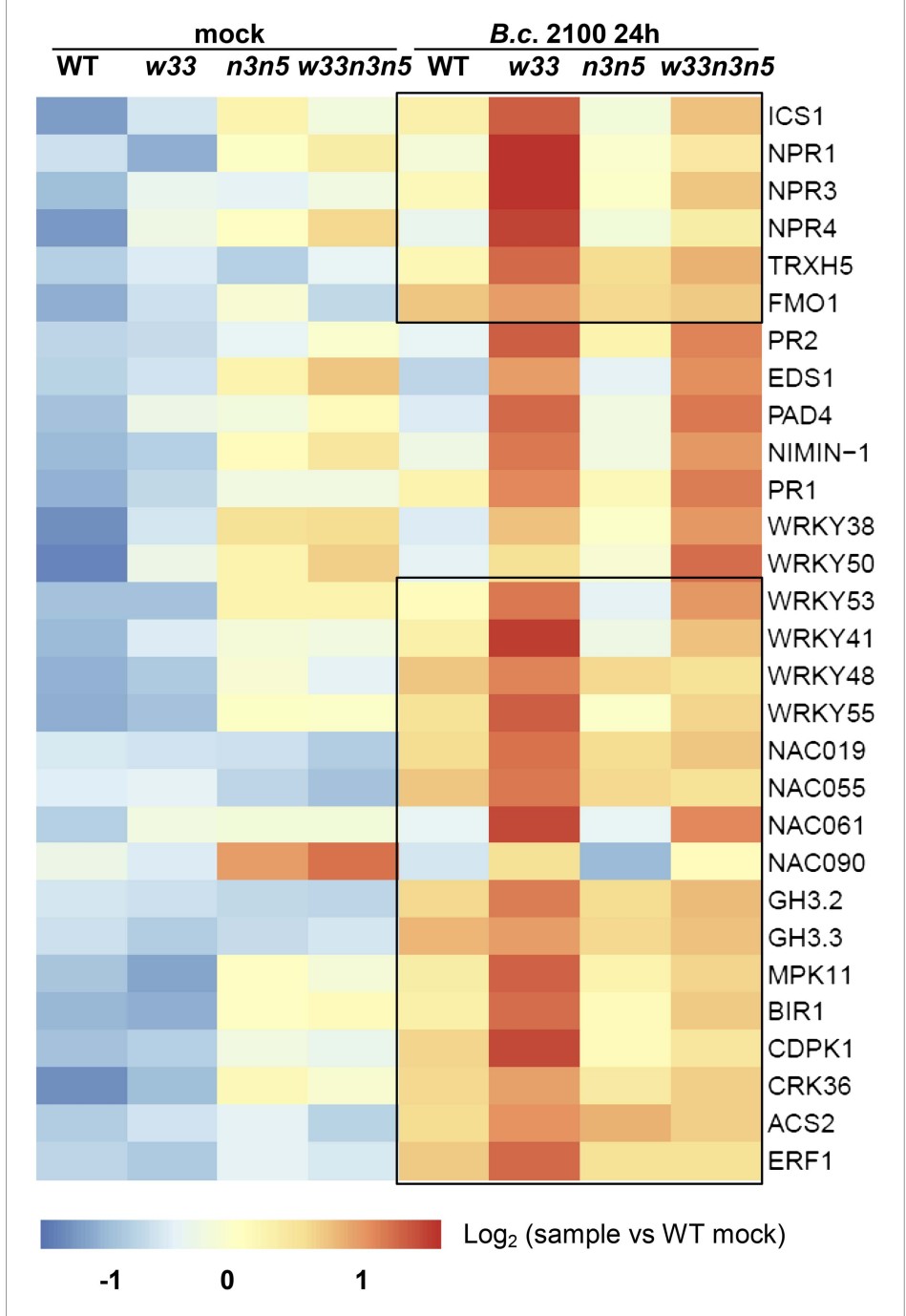

**Figure 7**. Expression of numerous genes up-regulated in infected *wrky33* plants showing WT-like levels in the *wrky33 nced3 nced5* triple mutant. Heatmap showing expression levels of genes, differentially expressed in RNA-seq and analyzed by qRT-PCR in WT, *wrky33* (*w33*), *nced3 nced5* (*n3n5*), and *wrky33 nced3 nced5* (*w33n3n5*) after mock treatment or 24 hr post *B. cinerea* (*B.c.* 2100) infection. Genes showing high expression levels in *wrky33* but reduced levels in the triple mutant are boxed. All values were normalized to the expression of At4G26410.

## Discussion

Signal transduction to the nucleus and a complex gene regulatory network governing the massive transcriptional reprogramming in the host upon pathogen perception drive the plant immune response. TFs are key terminal components of this signaling cascade and function by activating and

repressing the expression of numerous defense-associated genes. The TF WRKY33 plays a major role in conferring resistance of *Arabidopsis* plants to the fungal necrotroph *B. cinerea*. In the present work, combined genome-wide binding studies and transcriptional analyses allowed us to identify WRKY33 binding sites within the *Arabidopsis* genome upon fungal infection, and to correlate WRKY33 binding to altered transcriptional outputs. By including appropriate mutants in this study, we identified components of the ABA hormonal pathway that act downstream of WRKY33 to mediate host resistance.

Such a genome-wide analysis of in vivo target sites for a selected TF expressed under its native promoter in intact pathogen-infected plant tissue has not yet been reported. Thus, in a broader perspective, this study also sets the framework for establishing a comprehensive gene regulatory network model of plant immunity.

The number of high-affinity WRKY33 binding sites within the genome by far exceeds the number of direct WRKY33 target genes affected in their transcriptional response during *B. cinerea* infection. Overall, 80% of the WRKY33 targets were not significantly differentially expressed in the *wrky33* mutant upon infection compared to WT. This excess number of gene loci bound by a given TF but unaffected in their expression is consistent with previous ChIP-seq studies although the reasons for this discrepancy remain to be elucidated (*MacQuarrie et al., 2011*). One plausible explanation may be that transcriptional activation/repression at specific promoter sites is context dependent, and may require, apart from WRKY33 binding, additional diverse signaling inputs. For example, we detected strong enrichment of WRKY33 within the promoters of numerous genes coding for receptors of various MAMPs and damage-associated molecular patterns (DAMPs) including *FLS2*, *EFR*, *CERK1*, *PEPR1*, and *PEPR2*, but no altered expression of these genes in the *wrky33* mutant upon *Botrytis* infection. *WRKY33* is also strongly and rapidly induced during MAMP/DAMP-triggered immunity (*Lippok et al., 2007*; *Yamaguchi et al., 2010*), and thus a regulatory function of WRKY33 at these promoters may require additional signals/co-factors only triggered during MAMP/DAMP signaling. Additionally, spatial and temporal differences in the activation of target genes need to be considered. The current paper provides a snapshot of global WRKY33 function during *Botrytis* infection. Whether WRKY33 binding results in temporally distinct transcription patterns, as was observed for EIN3 in the ethylene response, remains to be investigated (*Chang et al., 2013*). However, such studies employing living pathogens on intact host plants remain challenging due to the asynchrony of the infection process at various cellular sites within the tissue.

Nevertheless, using conservative criteria for selecting differentially expressed genes in *wrky33* mutants compared to WT plants after *B. cinerea* 2100 infection, about 2600 genes were identified that showed transcriptional up- or down-regulation. The strikingly high number of modulated genes at early infection stages (14 hpi) highlights the importance of WRKY33 to initiate host responses to this pathogen.

## WRKY33 regulation of ABA signaling is critical for host defense towards *B. cinerea*

The role of ABA in biotic stress responses is complex and currently ill-defined. The ability of *Arabidopsis* to restrict penetration by the non-host barley pathogen *Blumeria graminis* was shown to be dependent on the NAC TF ATAF1-mediated repression of ABA biosynthesis (*Jensen et al., 2008*). In contrast, overexpression of *ATAF1* resulted in enhanced susceptibility of *Arabidopsis* plants to *B. cinerea* (*Wang et al., 2009*). ATAF1 was shown to directly bind to the *NCED3* promoter, which positively correlated with increased *NCED3* expression and ABA levels (*Jensen et al., 2013*). Moreover, transcriptomic studies using 4-week old detached *Arabidopsis* leaves infected with *B. cinerea* strain pepper revealed that genes involved in the suppression of ABA accumulation and signaling were up-regulated at early infection stages (*Windram et al., 2012*). Our study clearly demonstrates that increased expression of WRKY33 target genes associated with ABA biosynthesis (*NCED3* and *NCED5*) is causal for the susceptibility of *wrky33* to *B. cinerea* 2100, and the ABA deficient *wrky33 nced3 nced5* mutant restored WT-like resistance towards this necrotroph. Hence, our findings reveal a novel role of WRKY33 in modulating host resistance to *B. cinerea* by suppressing ABA accumulation/signaling (*Figure 8—figure supplement 1*). Interestingly, resistance to the necrotrophic fungus *Plectosphaerella cucumerina* is also negatively impacted by ABA, and *wrky33-1* mutant plants exhibited an enhanced susceptible phenotype towards this

pathogen (*Sánchez-Vallet et al., 2012*). Stimulating *NCED3* expression and ABA biosynthesis has also been described as an important virulence strategy employed by the hemi-biotroph *P. syringae* DC3000 in *Arabidopsis* (*de Torres-Zabala et al., 2007*). Virulence to this pathogen is strongly reduced in ABA mutants. Whether WRKY33 is involved in modulating ABA signaling during this host–bacterial interaction, and whether *P. syringae* DC3000 suppresses *WRKY33* expression is unknown.

In our experiments, the expression of several NAC TF genes associated with ABA regulation was affected upon *Botrytis* infection in a WRKY33-dependent manner. In particular, increased expression of *NAC002* (*ATAF1*), *NAC019*, *NAC055*, *NAC061*, *NAC068* (*NTM1*), and *NAC090* was observed in the *wrky33* mutant. Like *ATAF1*, transgenic *Arabidopsis* lines overexpressing *NAC019* or *NAC055* displayed enhanced susceptibility to *B. cinerea* (*Bu et al., 2008*). In contrast, the *nac019 nac055* double mutant showed increased resistance to *B. cinerea* compared with WT plants. ABA has been shown to induce *NAC019* and *NAC055* expression (*Jiang et al., 2009*; *Zheng et al., 2012*). Whether any of these NAC factors, apart from ATAF1, can also target the *NCED* genes and thereby enhance ABA biosynthesis in the *wrky33* mutant remains to be tested.

ABA can repress SA-, ET-, and JA/ET-dependent signaling but also positively affect some JA responses (*Asselbergh et al., 2008*; *Ton et al., 2009*). Our genetic and phytohormone studies showed that elevated ABA levels in the susceptible *wrky33* mutant resulted in concomitant increases in SA, JA, and ACC levels upon *B. cinerea* 2100 infection, implying a positive effect of ABA on these hormone signaling components. Increased SA levels per se in the *wrky33* mutant however do not contribute to susceptibility as *wrky33 sid2* double mutant plants are as susceptible to *B. cinerea* 2100 as the single mutant (*Birkenbihl et al., 2012*; *Figure 8—figure supplement 1*). Interestingly, concurrent increases in ABA, JA/ET and SA have also been observed in the interaction of *Arabidopsis* with *P. syringae* DC3000 and with the vascular oomycete pathogen *Pythium irregular* (*Adie et al., 2007*; *de Torres-Zabala et al., 2007*). In the case of *P. irregular,* however, host resistance correlated with high ABA levels, whereas ABA mutants were clearly more susceptible.

Our molecular analysis of *Botrytis*-challenged *wrky33* and *wrky33 nced3 nced5* plants confirmed that elevated ABA levels mainly activate NPR1-dependent SA signaling while not affecting the upstream EDS1-PAD4 pathway (*Figure 7*). Increased ABA also activated *ACS2*, *ACS6*, *ERF1*, and *ORA59*, targets involved in ET/JA signaling. In the *wrky33 nced3 nced5* mutant as in WT plants expression of *ACS6*, *ORA59,* but also *ERF5* and *PDF1.2* (data not shown) was significantly reduced. However, as both genotypes are resistant to *B. cinerea* these ET/JA components appear not to be essential in maintaining plant resistance.

How elevated ABA levels trigger activation of these hormone signaling cascades and specific TFs remains to be elucidated. The receptors for ABA are known (*Miyakawa et al., 2013*), but the molecular mechanisms linking downstream ABA signaling to the other hormonal pathways require further investigation. It is conceivable that the increased ABA levels in *wrky33* during *Botrytis* infection trigger the activation of currently unidentified downstream ABA-response factors that bind to the ABA response elements (ABRE, ACGTGG/T) or G-boxes (CACGTG) present in some gene promoters, resulting in transcriptional activation. Indeed, several genes including *NAC019*, *NAC061*, *FMO1*, *GH3.2*, and *GH3.3* contain such conserved motifs, which could respond to and be activated by ABA. In addition, the elevated levels of SA in *wrky33* during *Botrytis* infection may in part be responsible for the strong expression of the glutaredoxin gene *GRX480/ROXY19* observed in this mutant (RNA-seq data in this study; *Birkenbihl et al., 2012*). GRX480 binds to class II TGA factors, thereby activating TGA-regulated SA responses while preventing their participation in JA-mediated signaling (*Zander et al., 2014*). Plants ectopically overexpressing *GRX480* are susceptible to *B. cinerea* 2100 (*Birkenbihl et al., 2012*).

## WRKY33 is a dual functional transcription factor

In animals and humans, several TFs can act either as transcriptional activators or repressors, depending on DNA-binding sequences or interaction with additional co-factors (*Alexandre and Vincent, 2003*; *Berger and Dubreucq, 2012*; *Sakabe et al., 2012*; *Zhu et al., 2012*). Moreover, many human TFs function as repressors as often as they act as activators (*Cusanovich et al., 2014*). In plants, few factors with dual functions have been unequivocally characterized. In tomato, the transcriptional activator Pti4 can repress the expression of *PR10-a* by forming a complex with the SEBP repressor

(*Gonzalez-Lamothe et al., 2008*). In *Arabidopsis,* the TF WUSCHEL acts mainly as a repressor in stem cell regulation, but can function as an activator of *AGAMOUS(AG)* during floral patterning (*Ikeda et al., 2009*). Also WRKY proteins can act as activators or repressors, and selected family members in diverse plant species have been identified as key regulators in diverse plant processes (*Rushton et al., 2010*). WRKY53 can activate or repress the expression of genes, depending on the nature of the target promoter sequence (*Miao et al., 2004*). WRKY6 activates *PR1* expression while suppressing the expression of its own gene, and that of its closely related family member *WRKY42* (*Robatzek and Somssich, 2002*). Our data show that also WRKY33 is a bi-functional TF that can act as an activator or as a repressor in a promoter-context dependent manner (*Figure 8*). WRKY33 positively regulates genes involved in camalexin biosynthesis such as *CYP71A13* and *PAD3* by directly binding to their promoter regions (*Mao et al., 2011*; *Birkenbihl et al., 2012*). Our study

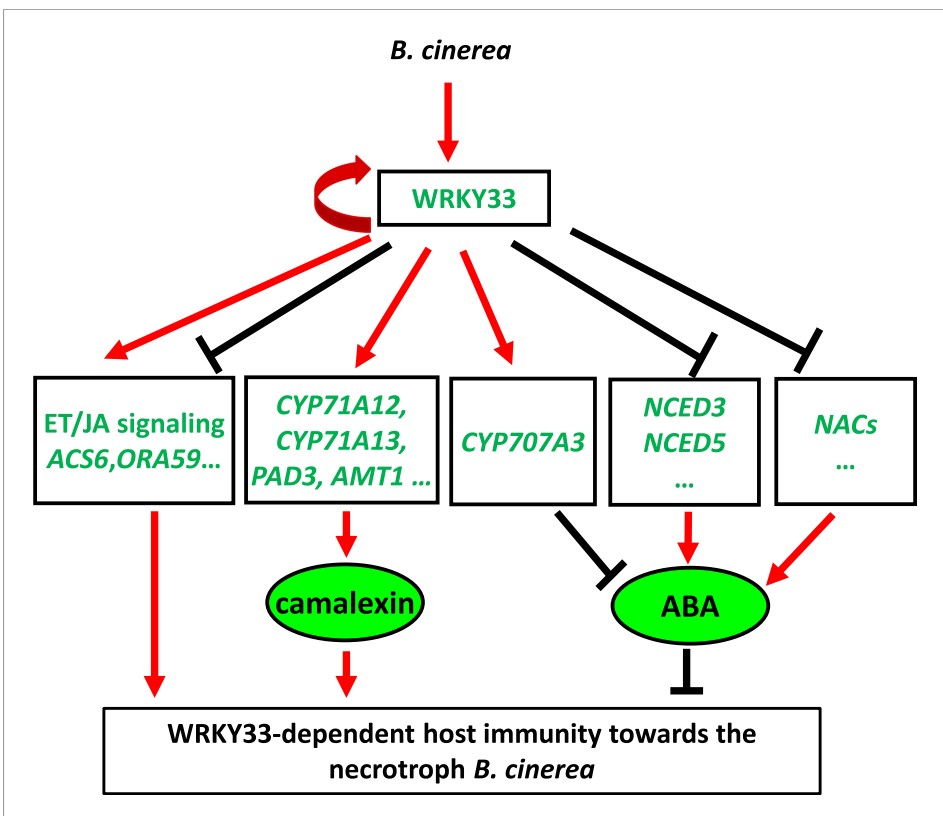

**Figure 8**. Dual regulatory role of WRKY33 in modulating host defenses to *B. cinerea* 2100. WRKY33 positively regulates target genes involved in camalexin biosynthesis thereby contributing to host resistance towards *B. cinerea* 2100. Target genes involved in ET/JA biosynthesis and signaling can either be positively or negatively regulated by WRKY33. On the other hand, WRKY33 negatively regulates ABA levels by directly targeting and repressing *NCED3* and *NCED5* expression, or inducing expression of *CYP707A3*, a gene involved in ABA metabolism. Thus, WRKY33 has both activator and repressor functions that may depend on promoter context. Red arrows indicate positive regulation, whereas black bars indicate negative regulation. The curved red arrow highlights positive feedback regulation of WRKY33 on its own gene promoter.

The following figure supplements are available for figure 8:

**Figure supplement 1**. Schematic representation showing WRKY33-dependent host immunity towards the necrotroph *B. cinerea* 2100 through repressing the ABA network.

**Figure supplement 2**. *AMT1* and *CYP71A12* expression is positively regulated by WRKY33.

**Figure supplement 3**. The role of *GH3* genes in WRKY33-mediated host resistance to *B. cinerea* 2100.

confirmed these observations and identified two additional camalexin biosynthetic genes, *AMT1* and *CYP71A12* that are positively and directly regulated by WRKY33 (*Figure 8—figure supplement 2*; *Supplementary file 2*). Mutants of *CYP71A13* and *PAD3* are susceptible to the necrotrophs *A. brassicicola* and *B. cinerea* (*Zhou et al., 1999*; *Nafisi et al., 2007*). Beyond this, several *Botrytis*-induced ET response genes were targeted and positively regulated by WRKY33 (*Figure 2E*). Still, WRKY33 had a negative regulatory relationship on the expression of >75% of all targets, implying that it mainly acts as a direct repressor of many defense genes following pathogen challenge.

How WRKY33 exerts its dual regulatory functions mechanistically requires further research. The simplest hypothesis would be that WRKY33 is recruited to distinct repressor and activator complexes at defined promoter sites. For instance, several WRKY33-interacting proteins containing a VQ motif have been discovered that influence defense gene expression (*Lai et al., 2011*; *Cheng et al., 2012*; *Pecher et al., 2014*). VQ proteins appear to act as suppressors of defense-related genes via their interaction with WRKY factors. Indeed, the protein VQ4/MVQ1 appears to function as a negative regulator of WRKY-type transcriptional activators including WRKY33. In a transient *Arabidopsis* protoplast assay, stimulation by MAMPs resulted in degradation of VQ4/MVQ1 following MAPK-mediated phosphorylation enabling WRKY33 to activate transcription of a defense-related reporter gene (*Pecher et al., 2014*). Two other proteins, SIB1 and SIB2, have been reported to interact with WRKY33 via their VQ motifs. This interaction was required to stimulate WRKY33 DNA-binding activity and very likely to positively regulate WRKY33-mediated resistance to necrotrophic fungi (*Lai et al., 2011*).

Interestingly, four VQ protein genes (*VQ8, VQ22/JAV1, VQ28, VQ33*) are also direct targets of WRKY33 and show altered expression upon *Botrytis* infection in *wrky33* compared to WT. VQ22/JAV1 functions as a negative regulator of JA-mediated defenses, and transgenic VQ22/JAV1 RNAi lines showed enhanced resistance to *B. cinerea* (*Hu et al., 2013*). Thus, elevated levels of these VQ proteins in *wrky33* may contribute to the suppression of JA signaling and susceptibility to this fungus.

Finally, many members of the WRKY TF gene family, including *WRKY33* itself, are also targets of WRKY33. The majority of *WRKY* genes are transcriptionally activated during immune signaling (*Pandey and Somssich, 2009*). Experimental and bioinformatics analyses have revealed that WRKY factors form a complex and highly interconnected regulatory sub-network that is positively and negatively affected by auto- and cross-regulation by various WRKY factors. This WRKY web appears to be deeply interconnected to various hormonal pathways at multiple levels, probably to ensure rapid and efficient signal amplification while allowing for tighter control in limiting the extent of the host immune response (*Eulgem, 2006*; *Llorca et al., 2014*).

In summary, genome-wide binding analysis and transcriptional profiling have identified potential targets of WRKY33, a key transcription factor involved in mediating resistance towards the necrotroph *B. cinerea* 2100. This study revealed that genes involved in ABA biosynthesis and directly regulated by WRKY33 act at crucial nodes in this signaling cascade. Due to the complexity of the highly interconnected hormonal signaling networks targeted by WRKY33, the precise molecular mechanisms underlying this resistance remain to be fully elucidated. In this respect, global transcriptional profiling of infected *wrky33 nced3 nced5* plants should prove extremely valuable to further narrow down key genes and sub-signaling pathways required to re-establish WT-like levels of resistance in this mutant.

## Materials and methods

### Plant material and growth conditions

For all experiments, *A. thaliana* ecotype Columbia (Col-0) was used. Besides WT, the following genotypes were employed: *wrky33* (GABI_324B11), *nced3-2, nced5-2, nced3 nced5, aba2-12, aba3-1, cyp707a1-1, cyp707a2-1, cyp707a3-1, wrky33 sid2-1, wrky33 npr1-1, wrky33 wrky70, gh3.2, gh3.3-1*. The double or triple mutants; *wrky33 nced3, wrky33 nced5, wrky33 nced3 nced5, gh3.2 gh3.3-1*, and *wrky33 gh3.2 gh3.3* were generated by crossing single or double mutants followed by PCR-based verification using appropriate primers (*Supplementary file 6*).

Plants were grown for 4 weeks under short-day conditions in closed cabinets (Schneijder chambers: 16 hr light/ 8 hr dark cycle at 22–24°C, 60% relative humidity) on 42 mm Jiffy-7 pots (Jiffy) to prevent

contaminations from garden soil. Before sewing, the Jiffy pot peat pellets were re-hydrated in water containing 0.1% liquid fertilizer Wuxal (Manna).

## Pathogen inoculation procedure

*B. cinerea* strain 2100 was cultivated on potato dextrose plates at 22°C for 10 days. Spores were collected, washed, and frozen at −80°C in 0.8% NaCl at a concentration of $10^7$ spores ml$^{-1}$. For inoculation of *Arabidopsis* plants, the spores were diluted in Vogel buffer prepared as previously described (*Birkenbihl et al., 2012*). For droplet inoculations, 2 μl of $2.5 \times 10^5$ spores ml$^{-1}$ were applied to single leaves of 4-week old intact plants. Leaves were excised from plants only for photographic documentation. The same spore concentration was used for spray inoculations of 4-week old intact plants. For mock treatment, Vogel buffer was used. Plants were kept prior to and during infection under sealed hoods at high humidity.

## ChIP-seq assay

4-week old WT plants or plants expressing *WRKY33-HA* from the native *WRKY33* promoter ($P_{WRKY33}$:*WRKY33-HA*) were spray inoculated or mock treated for 14 hr. ChIP assays were performed as previously described (*Birkenbihl et al., 2012*) following the modified protocol by *Gendrel et al. (2005)*, using rabbit polyclonal antibodies to HA (Sigma-Aldrich, St Louis, MO). ChIP DNA was purified using a QIA quick PCR Purification kit (Qiagen, Germany) and subjected to a linear DNA amplification (LinDA) protocol (*Shankaranarayanan et al., 2011*) which included two rounds of 'in vitro transcription' by T7 RNA polymerase. The resulting LinDA DNA was used to generate sequencing libraries bearing barcodes using a NEBNext ChIP-seq Library Pre Reagent Set for Illumina kit (New England Biolabs, Ipswich, MA). Sequencing was performed on Illumina HiSeq2500 at the Max Planck Genome Centre Cologne and resulted in about 10 million 100 bp single-end reads per sample. ChIP-qPCR validation of WRKY33 target genes was performed using gene specific primers (*Supplementary file 7*).

## ChIP-seq data analysis

Before mapping, remaining LinDA adapters and low quality sequences were removed from the sequencing data using a two-step procedure. In this procedure, first Bpm and t7-Bpm sites were trimmed from the 5′ end using cutadapt (version 1.2.1) (*Martin, 2011*) with options–e 0.2, -n 2 and–m 36 (otherwise default settings were used), and subsequently poly-A and poly-T tails and low quality ends were trimmed and reads with overall low quality or with less than 36 bases remaining after trimming were removed using PRINSEQ lite (version 0.20.2) (*Schmieder and Edwards, 2011*) with options–trim_qual_right/left 20, trim_tail_right/left 3 –min_len 36, -min_qual_mean 25. After this pre-processing steps, the remaining high quality reads were mapped to the *A. thaliana* reference genome TAIR10 (http://www.arabidopsis.org) using Bowtie (version 0.12.7) (*Langmead et al., 2009*) with options–best–m 1 to extract only uniquely mapped reads and allowing two mismatches in the first mapping steps (default settings). The ChIP-seq data sets used in this study have been deposited at the GEO repository (GSE66289).

To identify genomic DNA regions enriched in sequencing reads in the ChIP sample compared to input control as well as in inoculated compared to mock treated samples ('peak regions'), the peak calling algorithm of the QuEST program (version 2.4) (*Valouev et al., 2008*) was applied using the TF mode (option '2'), with permissive parameter settings for the peak calling (option '3'). Each of the two biological replicates was first analyzed separately and additionally, to obtain more exact peak locations for the consistent peaks, the mapped reads of the two replicates were pooled and peaks were also called for the pooled samples. To annotate the peak location with respect to annotated gene features in TAIR10 the annotatePeaks.pl function from the Homer suite (*Heinz et al., 2010*) was used with default settings. To extract consistent peaks between the replicates, a custom R (http://www.r-project.org) function (*Source code 1*) was used that identified overlapping peak regions between the replicates. Two peak regions were counted as overlapping, if they overlapped by at least 50% of the smaller region and a peak region was counted as consistent, if it was found to be overlapping between the two individual replicates as well as the pooled sample.

To search for conserved binding motifs in the consistent WRKY33 binding regions, for each consistent peak the 500 bp sequence surrounding the peak maximum was extracted and submitted to the online version of MEME-ChIP (*Machanick and Bailey, 2011*). MEME-ChIP was run with default settings, but a custom background model derived from the *Arabidopsis* genome was provided and 'Any number of repetitions' of a motif was allowed. For visualization, prominent motifs identified within MEME-ChIP by either MEME or DREME were chosen. To extract the number/percentage of peak regions that contain a certain motif, the online version of FIMO (*Grant et al., 2011*) was run with the peak sequences and the motif of interest (MEME/DREME output) as input and a p-value threshold of 0.001.

## RNA-seq assay

Total RNA was extracted from mock treated (14 hpi) and *B. cinerea* infected (14 hpi) 4-week old plants (WT and *wrky33*) using the RNeasy Plant Mini kit (Qiagen) according to the manufacturer's instructions, and mRNA sequencing libraries were constructed with barcodes using the TrueSeq RNA Sample Preparation Kit (Illumina). Three biological replicates were sequenced on Illumina HiSeq2500 by the Max Planck Genome Centre Cologne, resulting in 25–45 million 100 bp single end reads per sample. Total reads were mapped to the *Arabidopsis* genome (TAIR10) under consideration of exon-intron structures using the splice-aware read aligner TopHat (version 2.0.10) (*Kim et al., 2013*) with settings–a 10 –g 10 and known splice sites provided based on TAIR10 gene annotations. The RNA-seq data sets used in this study have been deposited at the GEO repository (GSE66290).

## Statistical analysis of RNA-seq

The mapped RNA-seq reads were transformed into a count per gene using the function coverageBed of the bedTools suite (*Quinlan and Hall, 2010*) with option–split to consider exon-intron structures. Genes with less than 50 reads in all samples together were discarded, and subsequently the count data of the remaining genes were TMM-normalized and log2-transformed using functions 'calcNormFactors' (R package EdgeR) (*Robinson et al., 2010*) and 'voom' (R package limma) (*Law et al., 2014*). To analyze differential gene expression between genotypes (WT, *wrky33*) and treatments (mock treated, *B. cinerea* infected), we fitted a linear model with the explanatory variable 'genotype_treatment' (i.e., including both genotype and treatment) using the function lmFit (R package limma). Subsequently, we performed moderated t-tests over the four contrasts of interest. Two contrasts compare *B. cinerea* infected vs mock treated samples within each genotype and the other two contrasts compare *wrky33* vs WT Col-0 plants within each treatment. In all cases, the resulting p values were adjusted for false discoveries due to multiple hypothesis testing via the Benjamini–Hochberg procedure. For each contrast, we extracted a set of significantly differentially expressed genes between the tested conditions (adj. p value $\leq$ 0.05, |log2FC| $\geq$ 1).

## Gene ontology analysis

GO term enrichment analysis on the gene sets of interest was performed using the R package goseq (*Young et al., 2010*) with custom GO term mappings obtained from org.At.tairGO2ALLTAIRS within the R package org.At.tair.db (*Carlson, 2010*). To identify enriched GO terms, the Wallenius distribution was used to approximate the null distribution and a probability weighting function was applied to correct for potential count biases in the analyzed gene sets. The resulting p values were adjusted for false discoveries due to multiple hypothesis testing via the Benjamini-Hochberg procedure and for each subset the significantly over-represented GO terms were extracted (adj. p value < 0.05). For the set of all WRKY33-regulated targets, the R package topGO (*Alexa and Rahnenführer, 2010*) was used to visualize the GO sub-graphs induced by the 10 most significantly enriched GO terms in the category 'Biological Process' and the five most significantly enriched GO terms in the category 'Molecular Function', respectively.

## qRT-PCR

Total RNA was isolated from leaves at 8, 14, 24, and 48 hpi with *B. cinerea* spores as described above and reverse transcribed with oligo(dT) primer to produce cDNA using the SuperScript First-Strand System for Reverse-Transcription PCR following the manufacturer's protocol (Invitrogen, Grand

Island, NY). cDNAs corresponding to 2.5 ng of total RNA were subjected to qPCR with gene-specific primers (*Supplementary file 8*) using the SYBR Green reagent (Bio-Rad, Hercules, CA). The qPCRs were performed on the iQ5 Multicolor Real-Time PCR Detection System (Bio-Rad) with three biological replicates. The relative expression was normalized to At4g26410 that was described as being highly constant under varying stress conditions (*Czechowski et al., 2005*). Data shown are means ± SD from the three biological replicates.

## Phytohormone measurements and quantification

Sample processing, data acquisition, instrumental setup, and calculations were performed as described (*Ziegler et al., 2014*). Instrument specific parameters for the detection of SA are shown in the Table below. $(3,4,5,6-D_4)$-SA was obtained from Campro Scientific (Veenendal, The Netherlands) and used as internal standard for SA quantification (1.5 ng per sample).

MS parameters for MRM-transition of salicylic acids (SA).

| Hormone | MRM transitions | Declustering potential (DP), V | Entrance potential (EP), V | Cell entrance potential (CEP), V | Collision potential (CE), V | Cell exit potential (CEX), V |
|---|---|---|---|---|---|---|
| SA | **137 → 93** | −25 | −5.5 | −14 | −22 | 0 |
| | *137 → 65* | −25 | −5.5 | −14 | −44 | 0 |
| SA-D$_4$ | **141 → 97** | −25 | −5.5 | −14 | −22 | 0 |

Quantifier and qualifier transitions are indicated in bold and italics, respectively.

## Quantification of fungal growth by qPCR

Quantification of fungal biomass relative to plant biomass by qPCR was basically performed as previously described (*Gachon and Saindrenan, 2004*). Leaves of the indicated *Arabidopsis* lines were inoculated with two 2 µl droplets of *B. cinerea* spores and DNA extracted 3 days later from whole leaves of similar fresh weight. The relative amounts of *B. cinerea* and *Arabidopsis* DNA were determined by qPCR employing specific primers for cutinase A and SKII, respectively.

## Expression of recombinant WRKY33 protein and EMSA

Full-length WRKY33 protein fused with NusA and an 8xHis-tag was expressed in the vector pMCSG48 in the *Escherichia coli* strain BL21 (DE3) Magic (kindly provided by Dr Michal Sikorski and Marta Grzechowiak, Institute of Bioorganic Chemistry, Poznan, Poland). Bacteria containing the *WRKY33* expression construct or the empty vector were induced with 0.5 mM isopropylthio-β-galactisude for 3 hr at 18°C. The His-tagged protein was purified using nickel affinity columns (QIAexpress Ni_NTA Fast Start, Qiagen, Germany) following the instructions of the manufacturer, and subsequently dialyzed against 20 mM Tris–HCl, pH 7, 5 overnight at 4°C.

The following DNA oligonucleotide probes were synthesized and biotin labeled by Sigma–Aldrich (Germany): W-box probe, 5′-CG<u>TTGACCG</u><u>TTGACC</u>GAG<u>TTGACT</u>TTTTA-3′; W-boxmut, 5′-CGTTGA<u>A</u>CGTTGA<u>A</u>CGAGTTGA<u>A</u>TTTTTA-3′; M-3, 5′-AA<u>TTTGAAT</u>AATCAAAGATCTTCC<u>TTTGAAT</u>TACCT<u>ATTCAAC</u>AT-3′ (derived from the *PROPEP3* promoter sequence); and M-7, 5′-GTCCACGCTTG<u>TTTGAAT</u>TTTCAGCCTTTGCAGGCAAGGT-3′ (derived from the *WAKL7* promoter sequence). Two complementary strands of the oligonucleotides were annealed by heating probes to 95°C for 3 min and then allowing probes to cool to room temperature overnight. Freshly prepared recombinant WRKY33 protein (1 µg) was incubated with the DNA probe (20 fmol) for 20 min at room temperature using the LightShift Chemiluminescent EMSA kit (Thermo Fisher, Germany) in the presence or absence of unlabeled competitor DNA. The resulting protein-DNA complexes were electrophoresed on 5% (wt/vol) polyacrylamide gels, and then transferred to N$^+$ nylon membranes in 0.5% Trisborate/EDTA buffer at 380 mA at 4°C for 60 min. Biotin labeled DNA detection was done according to the instructions of the manufacturer (Thermo Fisher). Bands were visualized using the BioRAD ChemiDoc MP Imaging System.

## Acknowledgements

We thank Dr Annie Marion-Poll (INRA, Versailles, France) for providing seeds of the *nced3*, *nced5*, and *nced3 nced5* mutants, Dr Catherine Bellini (Umeå University, Sweden) for the *gh3.3* seeds, Dr Kamal Bouarab (Université de Sherbrooke, Canada) for the *gh3.2* seeds, and Dr Wim Soppe (MPIPZ, Germany) for the *cyp707a1-1, cyp707a2-1, cyp707a3-1* seeds. The *E. coli* BL21 (DE3) Magic strain harboring the expression vector pMCSG48 containing the WRKY33 construct was kindly provided by Dr Michal Sikorski and Marta Grzechowiak (Institute of Bioorganic Chemistry, Poznan, Poland). We are grateful to Drs Paul Schulze-Lefert and Kenichi Tsuda (MPIPZ, Germany) for valuable comments and critical reading of the manuscript, and Elke Logemann for excellent technical assistance. This work was partly supported by an IMPRS PhD fellowship (SL) and by the Deutsche Forschungsgemeinschaft (DFG) grant SFB670 (IES and BK).

## Additional information

### Funding

| Funder | Grant reference | Author |
| --- | --- | --- |
| Max-Planck-Gesellschaft | IMPRS PhD fellowship | Shouan Liu |
| Deutsche Forschungsgemeinschaft | SFB 670 | Imre E Somssich |

The funders had no role in study design, data collection and interpretation, or the decision to submit the work for publication.

### Author contributions

SL, Conception and design, Acquisition of data, Analysis and interpretation of data, Drafting or revising the article; BK, Acquisition of data, Analysis and interpretation of data, Drafting or revising the article; JZ, Acquisition of data, Analysis and interpretation of data; RPB, IES, Conception and design, Analysis and interpretation of data, Drafting or revising the article

## Additional files

### Supplementary files

• Supplementary file 1. Summary of identified WRKY33 binding sites 14 hr post *B. cinerea* 2100 inoculation.

• Supplementary file 2. List of confirmed WRKY33 target genes.

• Supplementary file 3. List of WRKY33 regulated target genes involved in cell death.

• Supplementary file 4. List of WRKY33 regulated target genes associated in the GO category 'kinase activity'.

• Supplementary file 5. List of WRKY33 target genes encoding transcription factors.

• Supplementary file 6. List of primers used for genotyping.

• Supplementary file 7. List of primers used for ChIP-qPCR.

• Supplementary file 8. List of primers used for qRT-PCR.

• Source code 1. R function to identify overlapping peak regions.

## Major datasets

The following datasets were generated:

| Author(s) | Year | Dataset title | Dataset ID and/or URL | Database, license, and accessibility information |
|---|---|---|---|---|
| Liu S, Birkenbihl RP, Somssich IE, Kracher B | 2015 | WRKY33-dependent expression of Arabidopsis genes upon Botrytis cinerea 2100 inoculation | http://www.ncbi.nlm.nih.gov/geo/query/acc.cgi?acc=GSE66290 | Publicly available at the NCBI Gene Expression Omnibus (Accession no: GSE66290). |
| Liu S, Birkenbihl RP, Somssich IE, Kracher B | 2015 | WRKY33 binding sites in Arabidopsis upon Botrytis cinerea 2100 inoculation | http://www.ncbi.nlm.nih.gov/geo/query/acc.cgi?acc=GSE66289 | Publicly available at the NCBI Gene Expression Omnibus (Accession no: GSE66289). |
| Liu S, Birkenbihl RP, Somssich IE, Kracher B | 2015 | Analysis of WRKY33 binding sites and WRKY33-dependent gene expression in Arabidopsis thaliana upon Botrytis cinerea 2100 inoculation | http://www.ncbi.nlm.nih.gov/geo/query/acc.cgi?token=gnmxuwmqdlilfcz&acc=GSE66300 | Publicly available at the NCBI Gene Expression Omnibus (Accession no: GSE66300). |

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
