## [Decision Letter]

Thank you for sending your work entitled “Negative regulation of ABA signaling is critical for WRKY33-dependent *Arabidopsis* immunity towards *Botrytis cinerea*” for consideration at *eLife*. Your article has been in principle favorably evaluated by Detlef Weigel (Senior editor), a Reviewing editor, and two reviewers.

The Reviewing editor and the reviewers discussed their comments before we reached this decision, and the Reviewing editor has assembled the following comments to help you prepare a revised submission.

The reviewers agreed that your work identifying new WRKY33 targets during *Botrytis* infection is very interesting. The work nicely showed that you pinpointed new players in WRKY33-mediated resistance. Specifically, you reported that ABA (not SA) is a major player in WRKY33-activated resistance. While we are in principle supportive, there are some specific issues that need to be addressed to strengthen the work and acceptable for publication.

1) 1684 WRKY33 binding sites from genomic regions were associated with 1567 genes using ChIP-seq. Of them, 76% were W-box motifs which were consistent with previously reported WRKY family TF-binding sites. Moreover, you identified a new motif not previously known to bind WRKY factors, T/GTTGAAT. We think this is one of the major novelties of this study, but additional evidence needs to be provided to determine whether this is mediated by direct binding, or by other factors. If this motif is functionally as important as the W-box motif, the genomic regions containing such motifs should be undergoing evolutionary selection and their sequences should be conserved. We suggest that you compare the genomic regions containing W-box motif and/or T/GTTGAAT motif with the reported conserved DNA elements as described in Haudry et al. (2013)(PubMed ID: 23817568). Important WRKY33 target genes, such as *NCED* genes, would be expected to be associated evolutionarily conserved DNA elements.

2) For at least a couple of evolutionarily conserved T/GTTGAAT motifs, it is desirable to use a method like EMSA to test for direct binding. Alternatively, the authors might design artificial reporters with multiple tandem motif repeats alone or in combination with W boxes to test their regulatory activities (using constructs with single or double tandem motif repeats or truncated motifs as a control). This could be done as shown in a recent study (Li et al. Plant Cell 2015; PubMed ID: 25691733).

3) It has been known that TF-binding sites have dosage effects. For the TF-binding sites with positive regulation activities, multiple copies of motifs or binding signals on promoter regions are often associated with higher transcriptional induction levels. We suggest the authors to carry out a genome-wide correlation analysis between the binding signals/motif numbers and expression fold-changes for all the WRKY33 positively regulated target genes. This is feasible since the authors already have the ChIP-seq and RNA-seq datasets in hand.

4) Please clarify whether you normalized the gene expression levels on the basis of Per Kilobase of exon model per Million mapped reads (FPKM). Were exon-intron structures considered? If so, please describe this in the Materials and methods section. Otherwise, such expression level analysis may have bias on the genes with longer exons, which may further result in false positive detection during GO enrichment analysis. We suggest the authors use CuffDiff and/or DESeq2 to calculate Per Kilobase of exon model per Million mapped reads (FPKM) for all the TAIR0 genes and normalize their expression levels as described by Trapnell et al. 2012 (PubMed ID: 22383036).

---

## [Author Response]

*1) 1684 WRKY33 binding sites from genomic regions were associated with 1567 genes using ChIP-seq. Of them, 76% were W-box motifs which were consistent with previously reported WRKY family TF-binding sites. Moreover, you identified a new motif not previously known to bind WRKY factors, T/GTTGAAT. We think this is one of the major novelties of this study, but additional evidence needs to be provided to determine whether this is mediated by direct binding, or by other factors. If this motif is functionally as important as the W-box motif, the genomic regions containing such motifs should be undergoing evolutionary selection and their sequences should be conserved. We suggest that you compare the genomic regions containing W-box motif and/or T/GTTGAAT motif with the reported conserved DNA elements as described in Haudry et al. (2013)(PubMed ID: 23817568). Important WRKY33 target genes, such as* NCED *genes, would be expected to be associated evolutionarily conserved DNA elements*.

As suggested by the reviewers, we have checked whether the T/GTTGAAT motif is contained in the conserved noncoding sequences (CNS) described by Haudry et al. (2013) (PubMed ID: 23817568). The W-box motif has already been reported to be a conserved DNA element in that previous publication. Our analysis found that both the W-box and the T/GTTGAAT motifs that are located within our WRKY33 ChIP peak regions tend to be associated more strongly with CNS than the corresponding motifs outside of WRKY33 ChIP peaks. Still, overall the T/GTTGAAT motif is less common in the At genome, seems to be less enriched in CNS in general and also the association with CNS observed for motifs within WRKY33 ChIP peaks is clearly less pronounced than for the W-box. Taken together, we think that our exploratory analyses do not sufficiently support that this T/GTTGAAT motifs is strongly conserved and therefore undergoing evolutionary selection (if requested by the reviewers we can also supply the detailed results of our analyses). Thus, we feel at this stage we should not speculate too much on this point and therefore also are reluctant to include it in our manuscript.

In the course of re-analyzing our data realized there was an error in our previous Venn diagram (Figure 1—figure supplement 1). This error has no impact on our earlier conclusions but rather affect only absolute number of overlapping peaks.

*2) For at least a couple of evolutionarily conserved T/GTTGAAT motifs, it is desirable to use a method like EMSA to test for direct binding. Alternatively, the authors might design artificial reporters with multiple tandem motif repeats alone or in combination with W boxes to test their regulatory activities (using constructs with single or double tandem motif repeats or truncated motifs as a control). This could be done as shown in a recent study (Li et al. Plant Cell 2015; PubMed ID: 25691733)*.

We performed EMSA to address whether the newly identified element could be bound by WRKY33. For this we used as a positive control an oligonucleotide (designated W-box) containing 3 W-box motifs that has previously been shown by Mao et al. (Plant Cell 23, 1639-1653, 2011) to be bound by WRKY33 in EMSA, and as a negative control for binding specificity a similar oligonucleotide (designated W-boxmut), in which the TGAC core motif of the three W-boxes was mutated to TGAA. We synthesized two oligonucleotides containing the new motif (T/GTTGAAT) that were derived from two promoter sequences that we identified as WRKY33 targets in our ChIP-seq studies, and that resided within WRKY33 binding regions. One 45 bp oligonucleotide (designated M-3) was deduced from the *PROPEP3* gene promoter (At5g64905) and contained three copies of this new motif, and a second 40 bp oligonucleotide (designated M-7) derived from the *WAKL7* gene promoter (At1g16090), which contained one copy. WAKL7 was included because this promoter contains no W-box motifs within its entire promoter but still showed strong enrichment of WRKY33 in our ChIPseq and ChIP-qPCR assays. The results of this study are now included as Figure 1—figure supplement 2, and they clearly show that WRKY33 does not directly bind to the new motif. Moreover, this motif is also not capable of competing for W-box binding. Mention of these results was also added to the text (subsection headed “Genome-wide detection of *Arabidopsis* WRKY33 binding sites in response to *B. cinerea* 2100”) and description of the experiment was added to the Materials and methods section.

*3) It has been known that TF-binding sites have dosage effects. For the TF-binding sites with positive regulation activities, multiple copies of motifs or binding signals on promoter regions are often associated with higher transcriptional induction levels. We suggest the authors to carry out a genome-wide correlation analysis between the binding signals/motif numbers and expression fold-changes for all the WRKY33 positively regulated target genes. This is feasible since the authors already have the ChIP-seq and RNA-seq datasets in hand*.

As suggested by the reviewers we have performed genome-wide correlation analysis to test whether the number of the motifs (W-box, new motif, both motifs) present in a peak is correlated to the observed expression fold-changes of WRK33-induced target genes. This analysis did not reveal a clear positive correlation between motif number and gene induction (Spearman's rho between 0.071 and 0.273).

However, we are aware that only one *B. cinerea* induced time-point (14hpi) was selected for RNA-seq analysis. Thus, it is conceivable that the expression of various WRKY33 induced target genes have not yet reached or have already passed their expression maxima. We believe that without performing temporal expression studies no clear statement related to this point can be made.

*4) Please clarify whether you normalized the gene expression levels on the basis of Per Kilobase of exon model per Million mapped reads (FPKM). Were exon-intron structures considered? If so, please describe this in the Materials and methods section. Otherwise, such expression level analysis may have bias on the genes with longer exons, which may further result in false positive detection during GO enrichment analysis. We suggest the authors use CuffDiff and/or DESeq2 to calculate Per Kilobase of exon model per Million mapped reads (FPKM) for all the TAIR0 genes and normalize their expression levels as described by Trapnell et al. 2012 (PubMed ID: 22383036)*.

As the reviewers note, in RNA-seq analysis it is important to consider exon-intron structures. We have done so by using Tophat for the read alignment, which is a read-mapping tool that can map across and also identify splice junctions. And we also considered exon-intron structures when extracting the read counts per gene by using coverageBed with the option–split. As suggested by the reviewers we have now slightly rephrased the corresponding paragraphs in the Material and methods section to better clarify this.

In our subsequent differential expression analysis we have applied another accepted normalization approach, the so-called TMM (weighted trimmed mean of M-values [i.e. log fold changes]) normalization as proposed by Robinson and Oshlack (2010) (PubMed ID: 20196867). This normalization approach has been shown before e.g. by Dillies at al., 2012 (PubMed ID: 22988256) to be more efficient than the classic RPKM/FPKM normalization (at least when used without application of further scaling as done e.g. in newer versions of CuffDiff) and comparable to the normalization method used in DESeq/DESeq2. This TMM normalization was combined with a log2 transform to yield log2-counts per million (log2cpm) using the function ‘voom’ in the R package limma, which additionally estimates the mean-variance relationship for the log-counts and assigns a weight to each observation based on its predicted variance. These weights are then used in the subsequent linear modeling process to adjust for heteroscedasticity as described by [42] (PubMed ID: 24485249). This normalization procedure and subsequent linear modeling in limma (voom-limma) have been shown previously e.g. by Rapaport et al., 2013 (PubMed ID: 24020486) and Seyednasrollah et al., 2013 (PubMed ID: 24300110) to perform well at detecting differentially expressed genes on different datasets with respect to both sensitivity and specificity (also when compared with CuffDiff), and to be among the best tools when the number of replicates is relatively low, like with the three replicates we are analyzing.

We are aware that this approach also has its limitations, for example it considers expression at the gene level without regard to the potential isoform diversity and there might be certain specific cases where this gene-level analysis might not be able to correctly identify differential expression (for genes with both up-regulated and down-regulated isoforms). The suggested tool CuffDiff addresses this problem by combining the differential expression analysis with isoform deconvolution and is a reasonable and thus widely used alternative. However, we believe that the estimation of isoform-specific expression is a very difficult task, especially for genes with strongly overlapping isoforms and at moderate sequencing depths, and accordingly any isoform-specific expression estimates inevitably are associated with a certain degree of uncertainty. Overall, we think both approaches have their strengths and weaknesses, and, at least for the analysis of gene-level expression differences we are focusing on in this study, our workflow is just as much a reasonable approach as the suggested CuffDiff.

Nevertheless, to further verify the reliability of our results, we re-analyzed our RNA-seq data with both suggested alternative tools (DESeq2 and CuffDiff). A comparison of the gene sets identified with each tool as significantly differentially expressed (absolute log2FC ≥ 1, p < 0.05) for the four analyzed contrasts showed a large overlap between the gene sets. The vast majority of differentially expressed (DE) genes identified with our workflow (limma) were also identified as DE with DESeq2, and, to a slightly lesser extent, also with CuffDiff, with a major fraction of genes being consistently identified with all three tools (Figure 9). Similarly, also comparisons of the observed log2 fold changes between the tools showed an extremely high correlation between our workflow (limma) and DESeq2 (Figure 10) and still also a strong correlation between our workflow and CuffDiff (Figure 11) in spite of the conceptual differences between the tools. This strong agreement between the three tools in our view confirms the overall reliability of our results and supports our judgment that the workflow we used in our analysis is also a valid alternative to identify differentially expressed genes. As suggested by the reviewers, we have now rephrased the corresponding section in the Material and methods to describe the used approach in a bit more detail.

Author response image 1.Venn diagrams showing the overlap between the sets of genes identified as significantly differentially expressed (abs(log2FC) ≥ 1, p adj < 0.05) with the three different tools limma (used in our study), DESeq2 and Cuffdiff for the four analysed comparisons.**DOI:**
http://dx.doi.org/10.7554/eLife.07295.034

Author response image 2.Plots show the correlation between the log2 fold changes obtained for each gene with limma (used in our study) and the alternative tool DESeq2. Yellow points represent genes identified with limma as significantly DE (p adj < 0.05), blue circles represent genes identified with DESeq2, and an overlap of both marks represents genes consistently identified with both tools.**DOI:**
http://dx.doi.org/10.7554/eLife.07295.035

Author response image 3.Plots show the correlation between the log2 fold changes obtained for each gene with limma (used in our study) and the alternative tool Cuffdiff. Yellow points represent genes identified with limma as significantly DE (p adj < 0.05), blue circles represent genes identified with Cuffdiff, and an overlap of both marks represents genes consistently identified with both tools.**DOI:**
http://dx.doi.org/10.7554/eLife.07295.036

Still, it has been shown before, as the reviewers point out, that the detection of differentially expressed genes can have a bias on genes with longer exons or more general on genes with higher read counts. We also agree with the reviewers that the potential existence of such a bias would in turn result in a bias in the downstream GO term enrichment analysis. Thus, we explicitly checked our DE sets for the presence of such a bias, and an analysis of the gene length distributions did not reveal any evidence for a bias toward longer genes in our DE gene sets, also when compared to the DE gene sets obtained with the suggested alternative tool CuffDiff (Figure 12). However, the observed read counts in an RNA-seq experiment can be biased not only by differences in transcript length, but also by further factors like differences in sequence composition or secondary structure. Moreover, the gene set on which we actually performed the GO enrichment analysis contained only the subset of DE genes (wrky33 “regulated”) that were also found to be bound in our ChIP-seq (wrky33 “targets”), and it has been shown that also the identification of ChIP-seq targets can have a bias on more highly expressed genes. Thus, we additionally checked our specific gene sets of interest (“regulated”, “targets”, “regulated targets”) for the presence of an expression bias in general (Figure 13). As the ChIP-seq targets did indeed seem to have a certain bias towards higher expressed genes, we re-did the GO term enrichment analysis with a different tool (goseq) that can account for this potential expression bias in the analyzed gene set through the integration of a probability weighting function. In this way we can, within the GOterm analysis itself, correct for any expression bias in the analyzed gene set and thus reduce the chance of false positive detections. We have updated the corresponding text and figures (Figure 2, Figure 2—figure supplement 1) in our manuscript to now represent the results of this new GO term analysis with expression bias correction (which overall agree quite well with the previous analysis) and also adjusted the methods description accordingly.

Author response image 4.Boxplots represent the gene length distributions among the set of all analyzed genes and the subsets of genes identified as significantly differentially expressed (with abs(log2FC) ≥ 1, p adj < 0.05) in the four analyzed pairwise comparison with the voom-limma workflow used in our study (A) or the alterantive tool Cuffdiff (B).**DOI:**
http://dx.doi.org/10.7554/eLife.07295.037

Author response image 5.Boxplots represent the distributions of total read counts (over all samples together) per gene among the set of all analysed genes and the subsets of wrky33-regulated genes (from RNA-seq), wrky33 targets (from ChIP-seq) and wrky33-regulated wrky33 targets.**DOI:**
http://dx.doi.org/10.7554/eLife.07295.038